# DDX41 resolves G-quadruplexes to maintain erythroid genome integrity and prevent cGAS-mediated cell death

Honghao Bi[1,2], Kehan Ren [1,2], Pan Wang[1,2], Ermin Li[1,2], Xu Han[1,2], Wen Wang[3], Jing Yang[1,2], Inci Aydemir[1,2], Kara Tao[1], Renee Ma[1], Lucy A. Godley [2,4], Yan Liu [2,4], Vipul Shukla [2,5,6], Elizabeth T. Bartom [2,6,7], Yuefeng Tang[8], Lionel Blanc [8], Madina Sukhanova[1,2] & Peng Ji [1,2] ✉

Deleterious germline *DDX41* variants constitute the most common inherited predisposition disorder linked to myeloid neoplasms (MNs), yet their role in MNs remains unclear. Here we show that DDX41 is essential for erythropoiesis but dispensable for other hematopoietic lineages. Ddx41 knockout in early erythropoiesis is embryonically lethal, while knockout in late-stage terminal erythropoiesis allows mice to survive with normal blood counts. DDX41 deficiency induces a significant upregulation of G-quadruplexes (G4), which co-distribute with DDX41 on the erythroid genome. DDX41 directly binds to and resolves G4, which is significantly compromised in MN-associated *DDX41* mutants. G4 accumulation induces erythroid genome instability, ribosomal defects, and p53 upregulation. However, p53 deficiency does not rescue the embryonic death of Ddx41 hematopoietic-specific knockout mice. In parallel, genome instability also activates the cGas-Sting pathway, impairing survival, as cGas deficiency rescues the lethality of hematopoietic-specific Ddx41 knockout mice. This is supported by data from a DDX41-mutated MN patient and human iPSC-derived bone marrow organoids. Our study establishes DDX41 as a G4 resolvase, essential for erythroid genome stability and suppressing the cGAS-STING pathway.

Germline mutations in DDX41 predispose 2–5% of patients with myelodysplastic syndromes (MDS) or acute myeloid leukemia (AML)[1–6]. Patients with *DDX41* mutation tend to be late onset with long-term preceding indolent or mild cytopenia[4,7–9]. More than 80 distinct *DDX41* germline and somatic variants have been reported, making *DDX41* one of the most common MDS/AML predisposition genes[3]. DDX41 is an ATP-dependent DNA/RNA helicase and belongs to the DEAD box family of proteins[10]. It also binds to dsDNA to induce innate immune responses during viral infection[11,12]. Mouse and human DDX41 proteins are highly conserved. DDX41 is functionally important in multiple processes, including mRNA splicing, innate immunity, and rRNA processing[10]. Although clinical evidence shows clear pathologic roles

[1]Department of Pathology, Feinberg School of Medicine, Northwestern University, Chicago, IL, USA. [2]Robert H. Lurie Comprehensive Cancer Center, Northwestern University, Chicago, IL, USA. [3]Galter Health Sciences Library and Learning Center, Feinberg School of Medicine, Northwestern University, Chicago, IL, USA. [4]Division of Hematology and Oncology, Department of Medicine, Feinberg School of Medicine, Northwestern University, Chicago, IL, USA. [5]Department of Cell and Developmental Biology, Feinberg School of Medicine, Northwestern University, Chicago, IL, USA. [6]Center for Human Immunobiology, Feinberg School of Medicine, Northwestern University, Chicago, IL, USA. [7]Department of Biochemistry and Molecular Genetics, Feinberg School of Medicine, Northwestern University, Chicago, IL, USA. [8]Northwell Institute of Molecular Medicine, Feinstein Institutes for Medical Research, Manhasset, NY, USA. ✉e-mail: peng-ji@fsm.northwestern.edu

of *DDX41* germline and somatic mutations in myeloid neoplasms, often leading to DDX41 loss of function, the mechanism remains elusive. Previous research on DDX41 led to varied findings. A recent mouse genetic study revealed that biallelic *Ddx41* mutations disrupt snoRNA biogenesis and are incompatible with proliferating hematopoietic cells[13]. Conversely, another study showed that *ddx41* deficiency in zebrafish caused R-loop accumulation, resulting in the expansion of the hematopoietic stem and progenitor cells (HSPCs)[14]. Understanding DDX41's mechanisms across hematopoietic lineages is crucial for developing innovative therapies for DDX41-mutated MNs.

G-quadruplexes (G4) are four-stranded, noncanonical secondary DNA structures formed in guanine-rich sequences[15]. G4 was reported to be related to the R-loop formation[16,17]. Both structures show accumulation in G-rich regions and cause DNA double-strand breaks (DSBs) by inducing genome instability[14]. DEAD box family proteins, such as DDX5, were shown to be essential to resolve DNA G4[10,18]. G4 homeostasis is also critical for transcriptional regulation during development[19]. Through in vivo studies and human models, we show that DDX41 resolves G4 structures in the erythroid genome. DDX41 deficiency leads to G4 accumulation, causing genome instability, ribosomal defects, and cGAS-mediated cell death, impairing erythropoiesis and contributing to myeloid neoplasm pathogenesis.

## Results

### Ddx41 is essential for early-stage terminal erythropoiesis

Loss of Ddx41 in mouse hematopoietic cells in vivo was shown to be embryonically lethal with unclear cellular and molecular mechanisms[20]. To investigate Ddx41's functions in hematopoiesis, we first generated a hematopoietic-specific *Ddx41* knockout mouse model by crossing floxed *Ddx41* (*Ddx41^fl/fl*) with *VavCre* mice (Supplementary Fig. 1A). As reported, Ddx41 deficiency led to embryonic lethality[20], although VavCre expression in oocytes and endothelial cells raises the possibility that the observed lethality may involve contributions from non-hematopoietic lineages. Morphologic examination at E14.5 showed that *VavCre:Ddx41^fl/fl* embryo was severely pale, indicating defects in erythropoiesis (Supplementary Fig. 1B). We next purified Ter119 (a mature red cell marker) negative fetal liver HSPCs from the mutant fetuses and cultured the cells in an erythropoietin (Epo)-containing erythroid differentiation system[21,22]. We found that loss of Ddx41 completely blocked erythroid differentiation with a marked increase in cell death (Supplementary Fig. 1C and 1D), demonstrating that Ddx41 is essential for erythropoiesis.

To study the role of Ddx41 in erythropoiesis, we first determined the expression levels of Ddx41 during different stages of erythropoiesis in an in vitro mouse bone marrow erythroid culture system. We found that Ddx41 is highly upregulated in the early stages of terminal erythropoiesis (day 1 in culture) when the cells rapidly underwent differentiation and proliferation. Its level was then reduced on day 2 when the cells were prepared for enucleation (Fig. 1A). To study Ddx41 in vivo, we generated two erythroid-specific *Ddx41* knockout mouse models, *EpoRCre:Ddx41^fl/fl* and *HBBCre:Ddx41^fl/fl* mice, to test the functions of Ddx41 at different stages of erythropoiesis. *EpoRCre:Ddx41^fl/fl* mice manifest *Ddx41* deletion at the progenitor stages of erythropoiesis (approximately the CFU-E (colony forming unit-erythroid) stage, day 0 in culture system), whereas *HBBCre:Ddx41^fl/fl* mice exhibit *Ddx41* deletion at the terminal stages of erythroid differentiation[23] (Fig. 1B and Supplementary Fig. 1E). We found that *EpoRCre:Ddx41^fl/fl* mice were also embryonically lethal (Fig. 1C). The lack of appropriate morphogenesis suggests that Ddx41 could be critical for primitive erythropoiesis (Fig. 1D). We next analyzed *EpoRCre:Ddx41^fl/+* mice that survived to adulthood. These mice exhibited normal fetal liver erythropoiesis (Supplementary Fig. 2A). They showed mild macrocytic anemia with normal white blood cell and platelet counts at 2 months old (Fig. 1E and Supplementary Fig. 2B), which partially mimics patients with *DDX41* germline mutations before the development of overt MNs

upon somatic second hit[8,24,25]. We sacrificed the *EpoRCre:Ddx41^fl/+* mice and analyzed bone marrow erythroid cells using flow cytometry of cell surface CD71 and Ter119, two established markers for erythroid maturation. We found significantly compromised terminal erythropoiesis and increased myeloid cells percentage-wise and in absolute numbers with *Ddx41* heterozygosity (Fig. 1F and Supplementary Fig. 2C). This was accompanied by increased Ter119 positive erythroid cells in the spleen (Fig. 1G and Supplementary Fig. 2D), which indicates compensatory extramedullary stress erythropoiesis.

In contrast, *HBBCre:Ddx41^fl/fl* mice were viable with no detectable hematologic phenotypes, manifested by normal complete blood count (CBC) at 2 months old (Fig. 1H and Supplementary Fig. 2E). To confirm that *Ddx41* is knocked out at the late-stage terminal erythroblasts in *HBBCre:Ddx41^fl/fl* mice, we cultured *HBBCre:Ddx41^fl/fl* bone marrow lineage negative cells in the Epo-containing medium in which mouse erythroid progenitor cells can rapidly differentiate and proliferate to mature red blood cells in 2 days (Supplementary Fig. 2F). Western blotting demonstrated that Ddx41 showed a slight decrease on day 1 when the cells were predominantly at the early stages of terminal erythropoiesis proerythroblast stages[26–28]. Ddx41 was markedly decreased on day 2 when the cells were at the late-stage terminal erythropoiesis (Fig. 1I). Consistent with the CBC data, flow cytometry analyses showed no difference in the differentiation and enucleation of the cultured bone marrow erythroid cells from *HBBCre:Ddx41^fl/fl* and their littermate control mice (Fig. 1J). Similarly, we found no defects in erythropoiesis or the differentiation of other lineages in vivo in the bone marrow and spleen of *HBBCre:Ddx41^fl/fl* mice (Supplementary Fig. 2G-2K). Together, these results demonstrate that Ddx41 is critical at the early stages of erythropoiesis but dispensable at the late stages in mice.

We generated additional lineage-specific *Ddx41* knockout mouse lines to study its roles in hematopoiesis. These include *CD11c-Cre:Ddx41^fl/fl, LysMCre:Ddx41^fl/fl, and MRP8Cre:Ddx41^fl/fl*, which knockout *Ddx41* predominantly in dendritic cells, monocytes, and myeloid/granulocytes, respectively (Supplementary Table 1). Interestingly, these mice were all viable with no obvious abnormalities in their CBC at young ages. Flow cytometry studies revealed no obvious differences in the hematopoietic tissues in these animals (Supplementary Figs. 3–5), demonstrating that Ddx41 is dispensable in these lineages, particularly at their differentiated stages.

Given the significant role of Ddx41 in erythropoiesis in animal models, we wonder whether the same also occurs in MN patients with *DDX41* mutations. We identified an MDS patient carrying a somatic *DDX41* G1673 > A mutation (G530D) on the helicase domain, which was indicated to potentially disrupt the ATP binding to DDX41[5,8]. In addition to the *DDX41* mutation, this patient also carries a *PRPF8* T4916 > A somatic mutation. The patient has a long-standing MDS with anemia, neutropenia, and occasional mild thrombocytopenia. We purified the patient's bone marrow CD34+ HSPCs and CD71+ erythroid progenitor cells. Sanger sequencing showed that *DDX41* mutation exists in CD34+ cells but at a noise background level in CD71+ cells. In contrast, the *PRPF8* mutation is present at an equal allele frequency in both CD34+ and CD71+ cells (Fig. 1K). These findings suggest that *DDX41* mutation may impair erythroid differentiation or survival, resulting in the selective depletion of DDX41-mutant clones in the erythroid lineage.

### DDX41 controls the level of G-quadruplexes in early erythropoiesis

DDX41's role in erythropoiesis is underscored by studies in zebrafish, where its deficiency leads to anemia through ineffective erythropoiesis, involving DNA damage response pathways[29]. DDX41 is also critical in regulating DNA secondary structures such as R-loops[14,30], which frequently co-exist with G-quadruplexes (G4). G4 is known to stabilize R-Loop and regulate transcription by interacting with various

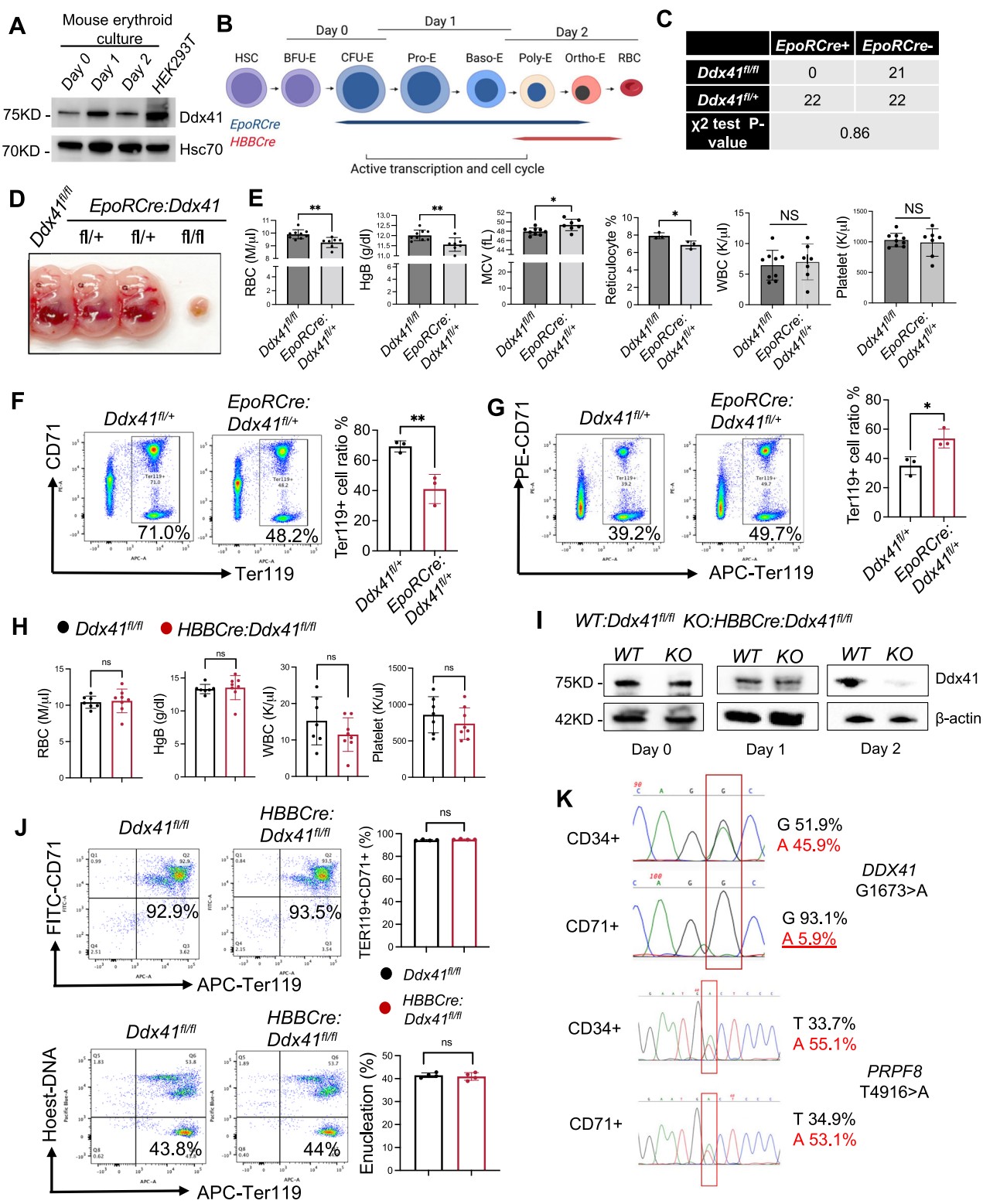

chromatin-binding proteins, including several DDX family proteins[15,19,31]. The role of G4 in erythropoiesis is unknown.

To study the role of G4 in erythropoiesis and whether the loss of DDX41 affects G4 formation, we first determined G4 levels in the bone marrow erythroid cells in vivo. We found that G4 is significantly increased in Ter119 positive erythroid cells compared to Ter119 negative cells (Fig. 2A). We next cultured mouse bone marrow lineage-negative cells in the Epo-containing medium and tested G4 levels at

different differentiation stages. We found that G4 significantly increased on day 1, corresponding to the highest proliferation and replication stage (Fig. 2B, and Supplementary Fig. 6A). The level of G4 decreased on day 2 when over 30% of the cells were enucleated. The same trend of changes in G4 levels was observed in cultured human CD34+ cells toward erythroid differentiation (Fig. 2C, and Supplementary Fig. 2F). Importantly, G4 levels in vivo in the erythroid cells are particularly higher than those in the other hematopoietic lineages

**Fig. 1 | Ddx41 is essential for early-stage terminal erythropoiesis. A** Western blotting of Ddx41 and Hsc70 (loading control) in the indicated cells. The loading control and Ddx41 were performed on the same gel (repeated 3 times). **B** Schematic illustration of erythropoiesis. Terminal erythropoiesis starts from the stages of CFU-E to proerythroblast (ProE). The stages of EpoR and HBB-driven Cre expression are shown. Created in BioRender. Ji, P. (https://BioRender.com/vycb0g0). **C** Mendelian segregation of indicated mice. A two-sided $\chi^2$ test yields a *P* value of 0.86, indicating no significant deviation from the expected Mendelian segregation. **D** Representative pictures of E14.5 embryos with the indicated genotypes. **E** Complete blood count (CBC) of the indicated mice at 2 months of age. From left to right, p values: 0.004, 0.0079, 0.0133, 0.032, 0.703, and 0.6381. **F** Flow cytometric assays of bone marrow erythroid cells from the indicated mice from E. Statistical analysis is on the right. *p* = 0.009. **G** Same as F except spleens were

analyzed. *p* = 0.022. **H** CBC analysis of indicated mice at 2 months of age. No statistically significant differences were found. **I** Western blotting of Ddx41 from the ex vivo Epo medium-cultured bone marrow erythroid cells from the indicated mice at 2 months of age. **J** Flow cytometric assays of bone marrow erythroid differentiation (upper panels) and enucleation (bottom panels) from the indicated mice in G. No statistically significant differences were found (right). **K** Sanger sequencing of *DDX41* and *PRPF8* in CD34+ HSPCs and CD71+ erythroid progenitor cells purified from bone marrow aspirate of a patient with MDS. *DDX41* mutation is somatic with unclear zygosity. All the error bars represent the SEM of the mean. At least 3 independent biological replicates were tested for each group. Each replicate represents either an individual mouse or a cell culture sample. The comparison between two groups was evaluated with 2 tailed t test. * *p* < 0.05, **p* < 0.01. ns: not significant. Source data are provided as a Source Data file.

(Figs. 2D, E), further indicating a critical role of G4 during erythropoiesis.

Since *VavCre:Ddx41^{fl/fl}* mice are embryonically lethal, we used *EpoRCre:Ddx41^{fl/+}* and *HBBCre:Ddx41^{fl/fl}* mice to investigate Ddx41's role in G4 accumulation in vivo. We first analyzed G4 levels in the bone marrow erythroid cells in *EpoRCre:Ddx41^{fl/+}* mice using a Ter119 and CD44-based gating strategy in which different stages of erythroid precursors can be separated based on CD44 expression (Fig. 2F)[32,33]. We found a mild but statistically significant increase in G4 level in the proerythroblast population (population I) in *EpoRCre:Ddx41^{fl/+}* mice compared to their wild-type counterparts (Fig. 2G). G4 levels in the late-stage orthochromatic erythroblasts (population V) showed no difference (Fig. 2H). In *HBBCre:Ddx41^{fl/fl}* mice, we did not find significant differences among different populations in the same assay, which is consistent with the lack of phenotypes in these mice. It is also difficult to pinpoint what developmental stage(s) G4 starts accumulating in vivo in these mice since most of the erythroid cells in the marrow are at the late stages of terminal erythropoiesis[34] when G4 level is reduced. Therefore, we purified lineage-negative HSPCs from these mice and cultured them in vitro in Epo-medium. As Ddx41 started to reduce on day 1 (Fig. 1H), the G4 level started to increase. The G4 level returned to the normal range on day 2, as expected (Fig. 2I, and Supplementary Fig. 6B). These results are consistent with the phenotypes of these mice and the adverse effect of increased G4 in early, but not late, stages of erythropoiesis. To further demonstrate the role of DDX41 in reducing G4 levels in human erythroid cells, we knocked out DDX41 through CRSPR/Cas9 in CD34+ human HSPCs using two different sgRNAs. Indeed, this led to a significant reduction in DDX41 protein levels (Fig. 2J) and a marked increase in G4 levels and cell death (Figs. 2K, L). Consistent with these model systems, we also observed increased G4 levels in the bone marrow mononuclear cells of the MN patient with DDX41 mutation (Fig. 2M).

We next applied pyridostatin (PDS), a selective G4 binding small molecule that stabilizes G4 and perturbs G4 homeostasis[35], in our in vitro erythroid differentiation system. PDS significantly increased G4 levels in cultured day 1 erythroid cells (Fig. 3A), induced dosage-dependent inhibition of erythroid differentiation, and increased cell death (Fig. 3B). Treatment of wild-type mice with a three-day high dose of PDS also significantly compromised bone marrow erythropoiesis in vivo (Fig. 3C), although no anemia was observed due to the long half-life of red blood cells. We next treated wild-type mice with PDS for 6 weeks through subcutaneously implanted pumps that chronically release the compound. Indeed, this treatment led to significant anemia (Fig. 3D). The white blood cell count was also reduced, primarily due to PDS-induced lymphopenia (Fig. 3D, E). As expected, chronic PDS treatment also led to ineffective terminal erythropoiesis and proportionally increased myeloid cells in the bone marrow (Fig. 3F, G).

We performed murine in vitro colony-forming unit (CFU) assays under different concentrations of PDS. We found that erythroid-related colonies, such as BFU-E and CFU-GEMM, decreased more

significantly in number than myeloid cell colonies (CFU-GM) (Fig. 3H). The same results were obtained when we performed CFU assays in human CD34+ cells (Fig. 3I). We also knocked out DDX41 using CRISPR/Cas9 in CD34+ cells. We found significantly reduced BFU-E and CFU-E colonies (Fig. 3J), indicating DDX41 could also be involved in stem cell commitment to the erythroid lineage. These results show that the erythroid lineage is more sensitive to G4 stresses, reinforcing our finding that DDX41 deficiency significantly impairs erythropoiesis.

## DDX41 is distributed at overlapping genomic loci as G4 and resolves G4

The genome distribution of G4 has been studied in different cell types and various species[36–38]. The genome-wide G4 localization in hematopoietic cells and whether DDX41 co-localizes with G4 are unclear. To understand this, we performed CUT&RUN assays in purified maturing mouse bone marrow Ter119+ erythroblasts and bone marrow lineage-negative HSPCs. We found a significant co-distribution of G4 and Ddx41 at the genome level both in HSPCs and erythroblasts. The total number of Ddx41 and G4 peaks increased markedly as cells differentiated into erythroid cells, even though the proportion of overlapping peaks remained relatively stable (Fig. 4A, B). This is consistent with the critical role of Ddx41 in erythropoiesis. Their distributions are mainly in the intergenic and intron regions, consistent with a previous report[39]. Notably, as cells differentiated from HSPCs to the erythroid lineage, both G4 and Ddx41 binding showed increased enrichment in the promoter and coding regions (Fig. 4A). Motif enrichment analysis revealed that Ddx41- and G4-bound regions are significantly enriched for binding motifs of key hematopoietic transcription factors, including *Runx1*, *Gfl1b*, *Gata3*, and *Myc*, in both HSPCs and erythroid cells. (Supplementary Fig. 7A). We further confirmed these genomic studies using a confocal immunofluorescence assay of G4 and Ddx41 in cultured mouse erythroblasts, which indeed showed their partial co-distribution (Fig. 4C). Computational predictions indicate that ribosomal DNAs (rDNAs) are highly enriched for putative quadruplex formation due to their repetitive DNA sequences[36,40]. Consistent with these reports, we found enrichment of G4 and Ddx41 on rDNAs in both HSPCs and erythroid cells (Supplementary Fig. 7B, C).

The co-occupation of DDX41 and G4 in the erythroid genome and the upregulation of G4 after DDX41 depletion indicate that DDX41 functionally maintains the level of G4. DEAD box family proteins, such as DDX5, were known to resolve DNA G4[10,18]. To determine whether DDX41 resolves G4, we first performed a pull-down assay using biotin-conjugated G4 oligos. The canonical G4 motif is Gm-Xn-Gm-Xo-Gm-Xp-Gm where each G-tract (Gm, m = 2–4) is separated by loops (Xn, Xo, and Xp), and n, o, and p are the combination of nucleotides of various lengths (up to 7)[31]. We designed three different G4 oligonucleotides. Three non-G4 oligonucleotides were used as controls. These oligonucleotides were annealed with biotinylated counterparts and then captured on streptavidin magnetic beads. They were subsequently incubated with lysates from cultured mouse erythroblasts in the

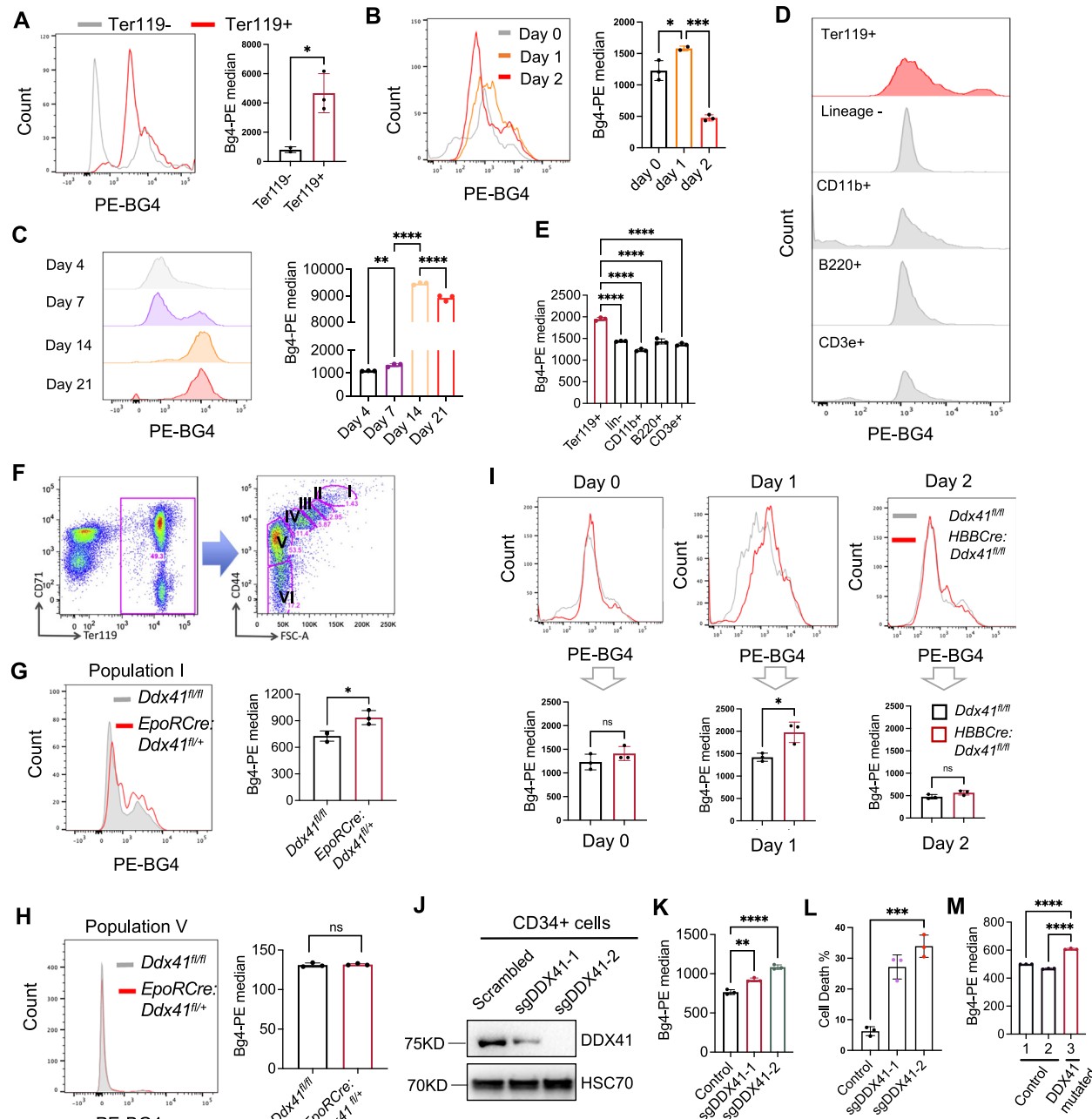

**Fig. 2 | Loss of DDX41 induces accumulation of G-quadruplexes. A** Flow cytometry assays of G4 levels from bone marrow Ter119-negative and positive cells. Quantification is on the right. *p* = 0.011. **B** Flow cytometry assays of G4 on the cultured mouse bone marrow lineage-negative cells in Epo medium. Quantification is on the right. *P* value left to right: 0.033, and 0.0002 **C** Flow cytometry assays of G4 from cultured human CD34+ HSPCs at the indicated time. Cells at day 7, 14, and 21 represent proerythroblasts, polychromatic to orthochromatic erythroblasts, and orthochromatic to mature red blood cells, respectively. *P* value from left to right: 0.002, < 0.0001, and < 0.0001. **D** Flow cytometric assays of G4 in the indicated mouse bone marrow lineage cells. **E** Quantification of D. *p* < 0.0001. **F** Gating strategy of various erythroblasts. Populations I to VI represent proerythroblasts, basophilic erythroblasts, polychromatic erythroblasts, orthochromatic erythroblasts, late orthochromatic to reticulocytes, and mature red blood cells, respectively. **G, H** Flow cytometry assay of G4 in bone marrow erythroid populations I (**G**) and V (**H**) from the indicated mice. Quantification is on the right. *P* value: G: 0.02, H: 0.71. **I** Flow cytometry assays of G4 on different days of cultured mouse bone marrow lineage-negative cells. Quantification is below the histogram. *P* value left to right: 0.98, 0.009, and 0.97. **J** CD34+ cells were transduced with lentiviral vectors expressing the indicated sgRNAs and Cas9. Cells were then harvested for Western blotting of the indicated proteins at day 9 in culture. **K** Quantitative analyses of G4 in cells from J using flow cytometric assays. *P* value from left to right: 0.0013, and < 0.0001. **L** Quantitative analyses of cell death in cells from J using flow cytometric assays. The dead cells are defined as propidium iodide and annexin V double positive. *p* = 0.0002. **M** Quantitative analyses of G4 in bone marrow mononuclear cells from the patient with DDX41-mutated MDS. All the error bars represent the SEM of the mean. The comparison between two groups was evaluated with 2 tailed t tests, and the comparison among multiple groups was evaluated with 1-way ANOVA tests. * *p* < 0.05, ***p* < 0.01, ****p* < 0.001, and *****p* < 0.0001. ns: not significant.

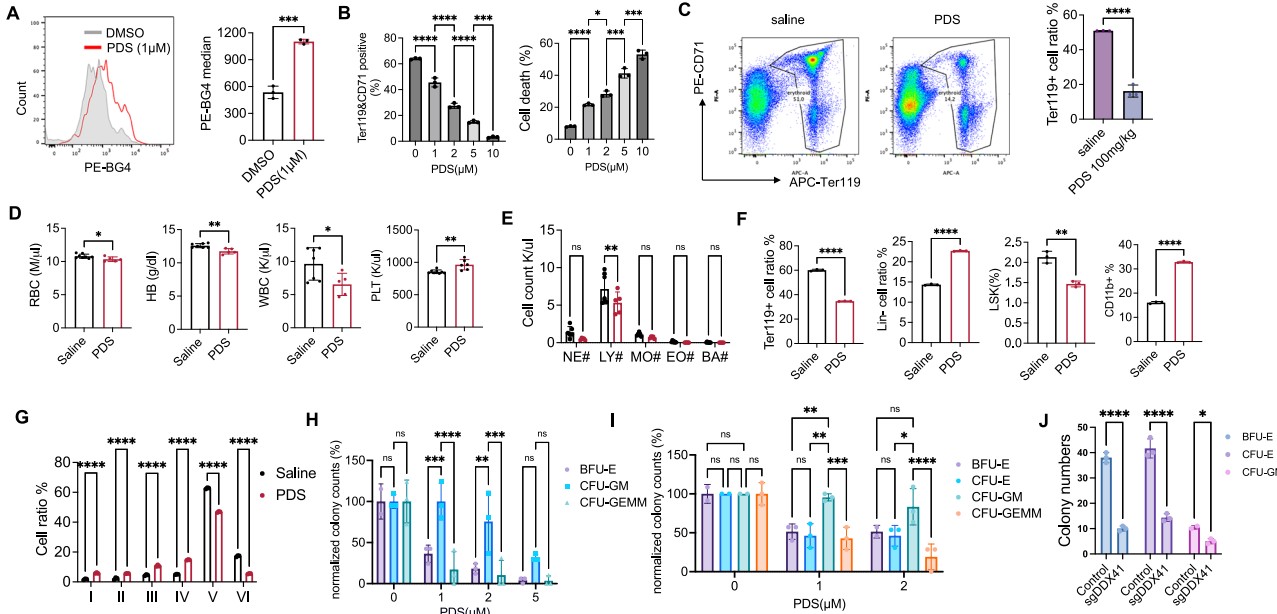

**Fig. 3 | G4 accumulation is detrimental to erythropoiesis. A** Bone marrow lineage-negative cells from 2-month-old mice were cultured for 24 h, treated with 1 μM PDS for 6 h, and analyzed by flow cytometry. Quantification is shown on the right. *p* = 0.0002. **B** Differentiation and apoptosis analyses of erythroid cells as in A with the indicated concentration of PDS. *P* value: *=0.015, ***< 0.001, ****< 0.0001. **C** Two-month-old wild-type mice received daily 100 mg/kg PDS injections for 3 days and were euthanized on day 3 for bone marrow erythroid analysis by flow cytometry. *p* < 0.0001. **D** Two-month-old wild-type mice were chronically treated with PDS or saline for 2 months, followed by a complete blood count. *P* value from left to right: 0.0345, 0.0021, 0.0357, and 0.0033. **E** Leukocyte count of (**D**). *P* value from left to right: 0.3839, 0.0031, 0.9518, >0.9999, and >0.9999. **F** Quantification of flow cytometry of different lineages and HSPCs in mice from (**D**). *P* value from left to right: < 0.0001, < 0.0001, 0.0021, and < 0.0001. **G** Erythroid cell analysis with CD44 and forward scatter in mice from (**D**). **H** Bone marrow lineage-negative cells, as in (**A**), were cultured with the indicated PDS in MethoCult M3434. Colonies were

quantified after 10 days. BFU-E: Burst-forming unit-erythrocyte; CFU-GM: Colony-forming unit-granulocyte-macrophage; CFU-GEMM: Colony-forming unit-granulocyte, erythrocyte, monocyte, and macrophage. Colony numbers are normalized to the untreated group. *P* value from left to right: > 0.9999, > 0.9999, 0.0005, < 0.0001, 0.0016, 0.0005, 0.1594, and 0.1505. **I** Human CD34+ cells were cultured in MethoCult H4435-enriched medium with the indicated concentration of PDS. Colonies were quantified after 14 days. CFU-E: Colony-forming unit-erythrocyte. *P* value left to right: 0.0071, 0.0062, 0.0005, 0267, and < 0.0001. **J** The same colony assay as (**I**), except that human CD34+ cells were transduced with Cas9 and DDX41 sgRNA. Control: CRISPR V2 vector with scrambled sgRNA. *P* value from left to right: < 0.0001, < 0.0001, and 0.039. All the error bars represent the SEM of the mean. The comparison between two groups was evaluated with 2-tailed t tests, and the comparison among multiple groups was evaluated with 1-way ANOVA tests. * *p* < 0.05, **p* < 0.01, ***p* < 0.001, and ****p* < 0.0001. ns: not significant.

presence of K+ cations. Ddx41 exhibited specific binding to all three G4 structures of distinct topologies (Fig. 4D), indicating its ability to recognize a wide range of G4 structures.

To determine whether DDX41 directly resolves G4, we performed an in vitro fluorescence resonance energy transfer (FRET) assay in which a fragment of G4 DNA is flanked by 6-fluorescein (6-FAM) on the 3′-end and black hole-1 quencher on the 5′-end[18] (Fig. 4E). Recombinant human DDX41 was added in vitro, together with G4-FRET and K+ cations. Indeed, we found a dose-dependent increase in the 6-FAM fluorescence with the increasing amount of DDX41, demonstrating that G4 was resolved by DDX41 (Fig. 4F, and Supplementary Fig. 8A). The kinetics of DDX41 in resolving G4 was also rapid and dose-dependent (Fig. 4G, and Supplementary Fig. 8B). Notably, in these experiments, the oligonucleotide was pre-folded into a G4 structure under high-K+ conditions and then incubated with recombinant human DDX41 in a low-K+ unfolding buffer, which limits the spontaneous refolding of G4. As a control, we included oligonucleotides processed identically but without the initial folding step. These unfolded oligos showed persistent fluorescence that remained stable under the assay conditions, confirming that refolding does not occur spontaneously during the assay (Supplementary Fig. 8C). We next tested two of the most common somatic mutations of *DDX41*, R525H, and G530D, on their influences on G4 binding capacities. We found that R525H mildly compromised the binding, whereas G530D markedly reduced it (Fig. 4H). We further discovered that both mutants lost G4 resolving activity in all three G4s we tested (Fig. 4I). These data

establish DDX41 as a G4 resolvase, which is compromised by its loss of function mutations.

DDX41 was reported to bind to various nucleic acid structures, including dsDNA, R-loop, and DNA/RNA hybrids[12,30,41]. To determine the relative affinity of DDX41 to these molecules, we performed a fluorescence polarization assay using recombinant DDX41. We found that G4 DNA represents a preferred or high-affinity substrate for DDX41 compared to other molecules (Fig. 4J), indicating that DDX41 may have evolved a specialized role in resolving G-quadruplexes.

## DDX41 deficiency-mediated G4 accumulation leads to genome instability, ribosomal biogenesis defects, and upregulation of p53

Accumulation of G4 is associated with genome instability and defects in ribosome biogenesis due to G4 enrichment on rDNAs[36,40,42–45]. Compromised ribosome biogenesis during erythropoiesis is well known to trigger p53-mediated cell death[46], which is believed to be the pathogenesis of Diamond-Blackfan anemia (DBA) and contribute to the development of del(5q) MDS[47,48]. We found increased γ-H2AX (a marker for genome instability) when the cultured mouse bone marrow HSPCs were treated with PDS (Fig. 5A–C). As expected, the transcription of ribosomal RNAs was significantly reduced (Fig. 5D). Defects in ribosomal RNA biogenesis are known to negatively influence the expression of ribosomal proteins[47,49]. Consistently, we found PDS treatment significantly reduced many ribosomal proteins, including Rps19 (mutated in 25% of DBA patients), Rps14 (haploinsufficiency in

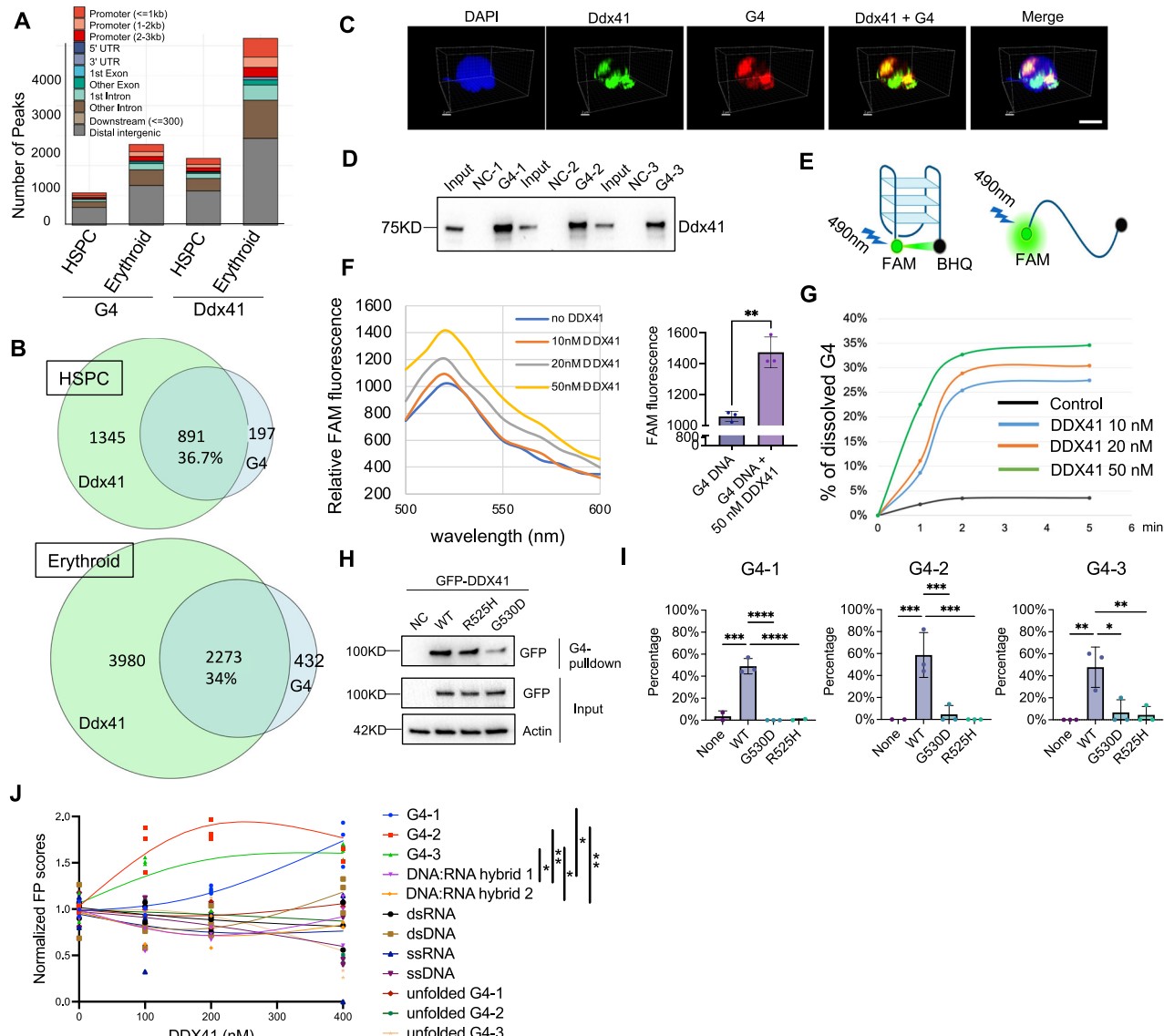

**Fig. 4 | DDX41 binds to and resolves G-quadruplexes. A** Genome-wide G4 and Ddx41 binding sites in mouse lineage-negative HSPCs and Ter119+ cells through CUT&RUN. **B** Quantification of the co-distributed genomic loci from (**A**). **C** Confocal immunofluorescence of G4 and Ddx41 in mouse lineage-negative cells after 1 day in Epo medium. Scale bar: 5 μm. The pictures are representative of >10 fields. **D** Mouse lineage-negative cells were cultured in Epo medium for 1 day. The lysates were incubated with indicated G4 or non-G4 control (NC) oligonucleotides in the presence of KCl for 1 h. Western blotting of Ddx41 was then performed. Data are representative of 3 independent experiments. **E** Schematic illustration of the FRET assay. Created in BioRender. Ji, P. (https://BioRender.com/ntess0a). **F** Dose-dependent increase of FAM signal with increased recombinant human DDX41 protein. Statistical analysis of FAM signals with 50 nM DDX41 compared to the G4 DNA-only group is on the right. $p = 0.0023$. **G** Time course of DDX41-mediated G4 resolving activity. **H** Western blotting analyses of indicated proteins in the G4 pull-down assay using biotin-conjugated G4 or non-G4 sequences and lysates from

HEK293T cells that were transfected with GFP-tagged wild-type or mutant DDX41. Data are representative of 3 independent experiments. **I** FRET assays using G4s, as in (**F**), incubated with wild-type or mutant recombinant DDX41. The Y-axis represents the percentage of unfolded G4 as in (**G**). All the error bars represent the SEM of the mean. $P$ value left to right: < 0.0001, < 0.0001, < 0.0001, 0.0002, 0.0003, 0.0002, 0.0042, 0.0101, and 0.0075. **J** Fluorescence polarization (FP) assay demonstrating preferential binding of DDX41 to G4 compared to other nucleic acid conformations. Oligonucleotides (20 nM) were incubated with increasing concentrations of DDX41 protein, and FP values were normalized to the FP signal in the absence of DDX41 (0 nM). G4-1 vs DNA:RNA hybrids ($p = 0.0101, 0.0047$); G4-2 vs DNA:RNA hybrids ($p = 0.0069, 0.0032$); and G4-3 vs DNA:RNA hybrids ($p = 0.0230, 0.0112$). The comparison between two groups was evaluated with 2-tailed t tests, and the comparison among multiple groups was evaluated with 1-way ANOVA. * $p < 0.05$, ** $p < 0.01$, *** $p < 0.001$, and **** $p < 0.0001$.

del(5q) MDS patients), and Rpl26 (mutated in certain DBA patients)[49]. The level of p53 was also increased (Fig. 5E). To directly investigate how DDX41 deficiency affects ribosomal biogenesis in vivo, we used *HBBCre:Ddx41^{fl/fl}* mice since these mice survive and Ddx41 is depleted in the late-stage erythroblasts when the cells are most abundant. We cultured the bone marrow lineage negative HSPCs from these mice in Epo-medium and found significantly decreased transcription of rRNAs on day 1 when Ddx41 starts to reduce, and G4 is significantly increased

(Fig. 5F, Fig. 2I). The ribosomal protein levels remain steady on day 1, possibly due to the stability of proteins compared to RNAs, but significantly reduced on day 2 (Fig. 5G). Interestingly, we found no increase in γ-H2AX when Ter119-negative erythroblasts from *HBBCre:Ddx41^{fl/fl}* or *EpoRCre:Ddx41^{fl/+}* mice were cultured in vitro (Supplementary Fig. 8D). These results indicate that these lineage-negative cells may be adapted to the Ddx41 deficiency during development.

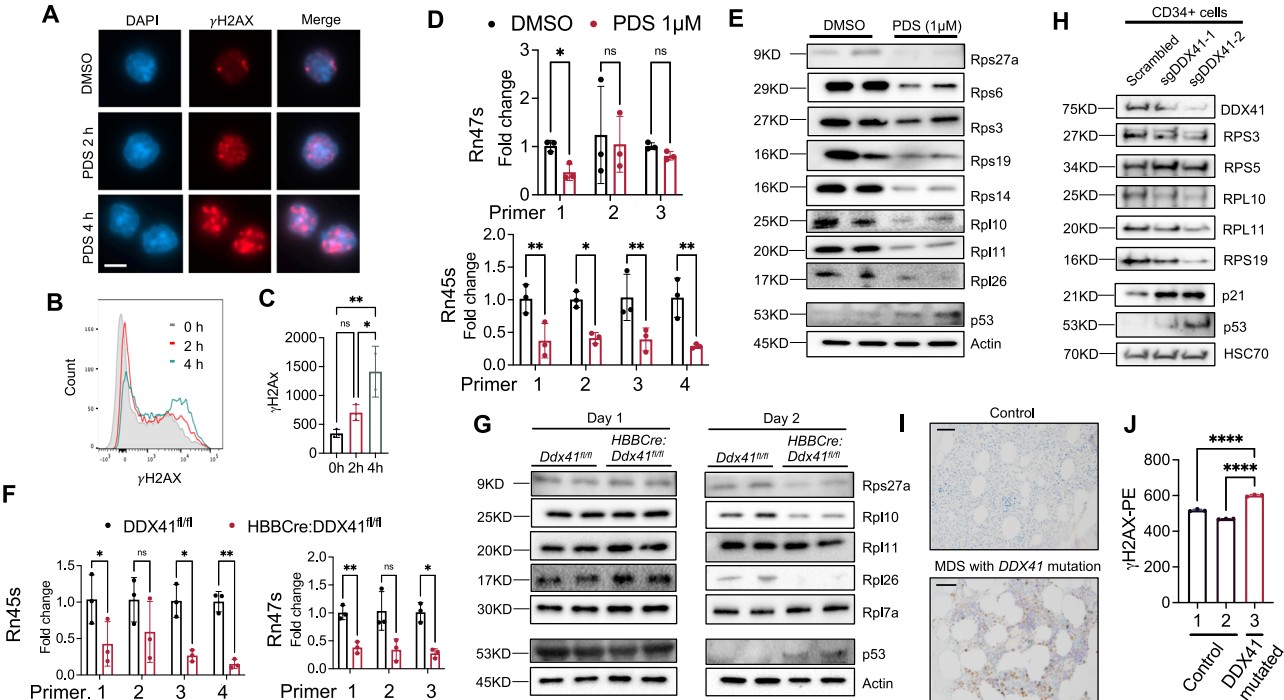

**Fig. 5 | DDX41 deficiency-mediated G4 accumulation leads to genome instability, ribosomal biogenesis defects, and upregulation of p53. A** Epo medium-cultured mouse bone marrow lineage negative HSPCs were treated with 1μM PDS for the indicated time. Immunofluorescence assays of γ-H2AX were performed, and representative images of the erythroid cells were presented. Scale bar: 5 μm. **B** Flow cytometry assay of the cells in (**A**). Data are representative of 3 independent experiments. **C** Statistical quantification of γH2AX signals in (**B**). 0 vs 2 h: $p = 2055$; 2 vs 4 h: $p = 0.0369$; 0 vs 4 h: $p = 0071$. **D** Epo medium-cultured mouse bone marrow lineage negative HSPCs were cultured for 1 day, followed by the treatment of 1μM PDS for 6 h. Quantitative RT-PCR analyses of indicated ribosome RNAs were performed using different primer sets. *P* value from left to right: Rn45s: 0.0100, 0.0181, 0.0100, and 0.0031; Rn47s: 0.0412, 0.9908, and 0.1099. **E** Western blotting assays of indicated in cells from (**D**). Actin is the loading control. **F** Same as (**D**) except that bone marrow lineage negative HSPCs from *HBBCre:Ddx41*$^{fl/fl}$ mouse

were cultured for 1 day before the quantitative RT-PCR assays. *P* value from left to right: Rn45s: 0.0462, 0.2082, 0.0119, 0.0042; Rn47s: 0.00307, 0.05250, and 0.00815. **G** Western blotting assays of the indicated proteins in (**F**). Cells from both day 1 and day 2 cultured cells were analyzed. Data are representative of 2 independent experiments. **H** CD34+ cells were transduced with lentiviral vectors expressing indicated sgRNAs and Cas9. Cells were then harvested for Western blotting of the indicated proteins at day 9 in culture. Data are representative of 2 independent experiments. **I** Immunohistochemical stains of p53 in bone marrow core biopsies from the patient in Fig. 1J and a normal individual. Scale bar: 100 μm. **J** Quantification of γ-H2AX in bone marrow mononuclear cells from the patient in (**I**) and 2 control individuals. All the error bars represent the SEM of the mean. The comparison between two groups was evaluated with 2-tailed t tests, and the comparison among multiple groups was evaluated using 1-way ANOVA tests. * $p < 0.05$, **$p < 0.01$, **** $p < 0.0001$, ns: not significant.

Consistent with these findings in mouse erythroid cells, DDX41 knockout in CD34+ HSPCs led to a similar reduction of various ribosomal proteins and upregulation of p53. The p53 downstream target p21 was also increased (Fig. 5H). We next tested whether the p53 level increased in the patient with *DDX41* mutation. We used bone marrow biopsies from the same patient with the G530D mutation and performed an immunohistochemical stain for p53. We found that the p53 level (Fig. 5I) and γ-H2AX (Fig. 5J) significantly increased in the patient with *DDX41* mutation compared to the normal control, consistent with the findings in the mouse models.

**Activation of the cGAS-STING pathway, but not p53 upregulation, is essential for DDX41 deficiency-mediated ineffective erythropoiesis**

While p53 mediates many pathologies of red cell-related diseases, it has been documented that overexpression of p53 does not induce overt abnormalities in a transgenic model[50]. To determine whether p53 mediates the major phenotypes of Ddx41 hematopoietic specific deficiency mice as it does in DBA and del(5q) MDS, we took a genetic approach and crossed p53 knockout mice with *VavCre:Ddx41*$^{fl/+}$ mice. If p53 is critical, p53 deficiency would rescue the lethality of *VavCre:Ddx41*$^{fl/fl}$ mice. However, no surviving *VavCre:Ddx41*$^{fl/fl}$ *Trp53*$^{-/-}$ mice were born (Fig. 6A, B). Dissection of the pregnant mice revealed that *VavCre:Ddx41*$^{fl/fl}$ *Trp53*$^{-/-}$ embryos

remained pale with an underdeveloped fetal liver (Fig. 6A). These results demonstrate that p53 is not essential to mediate ineffective erythropoiesis and cell death in the Ddx41 hematopoietic-specific knockout mouse model.

DDX41 was reported to sense intracellular dsDNA and activate STING to mediate type I interferon response in dendritic cells[12,51]. Indeed, we found the level of cGAMP, intracellular second messenger in response to cGAS activation, was significantly increased in PDS-treated erythroid cells (Fig. 6C). The downstream targets of the cGAS-STING pathway, including interferon beta (IFN-β) and the NF-κB signaling, were also activated (Fig. 6D, E). We previously revealed that erythroid cells generate transient nuclear openings in the early stages of terminal erythropoiesis[52,53], which could further activate cGAS. To test this, we treated the cultured mouse bone marrow erythroid cells with a caspase inhibitor, which blocks nuclear opening. This led to a significant reduction of cGAMP (Fig. 6F), suggesting that nuclear openings contribute to the vulnerability of the erythroid cells to the genome instability induced by G4 upregulation.

We then took a similar genetic approach and crossed cGas knockout mice with *VavCre:Ddx41*$^{fl/+}$ mice. Intriguingly, the cGas deficiency completely rescued the embryonic lethality of the *VavCre:Ddx41*$^{fl/fl}$ mice. *VavCre:Ddx41*$^{fl/fl}$cGas$^{-/-}$ (DKO) mice showed no evidence of anemia or other cytopenias (Fig. 6G, and Supplementary Fig. 9A, B). The bone marrow hematopoiesis was also intact

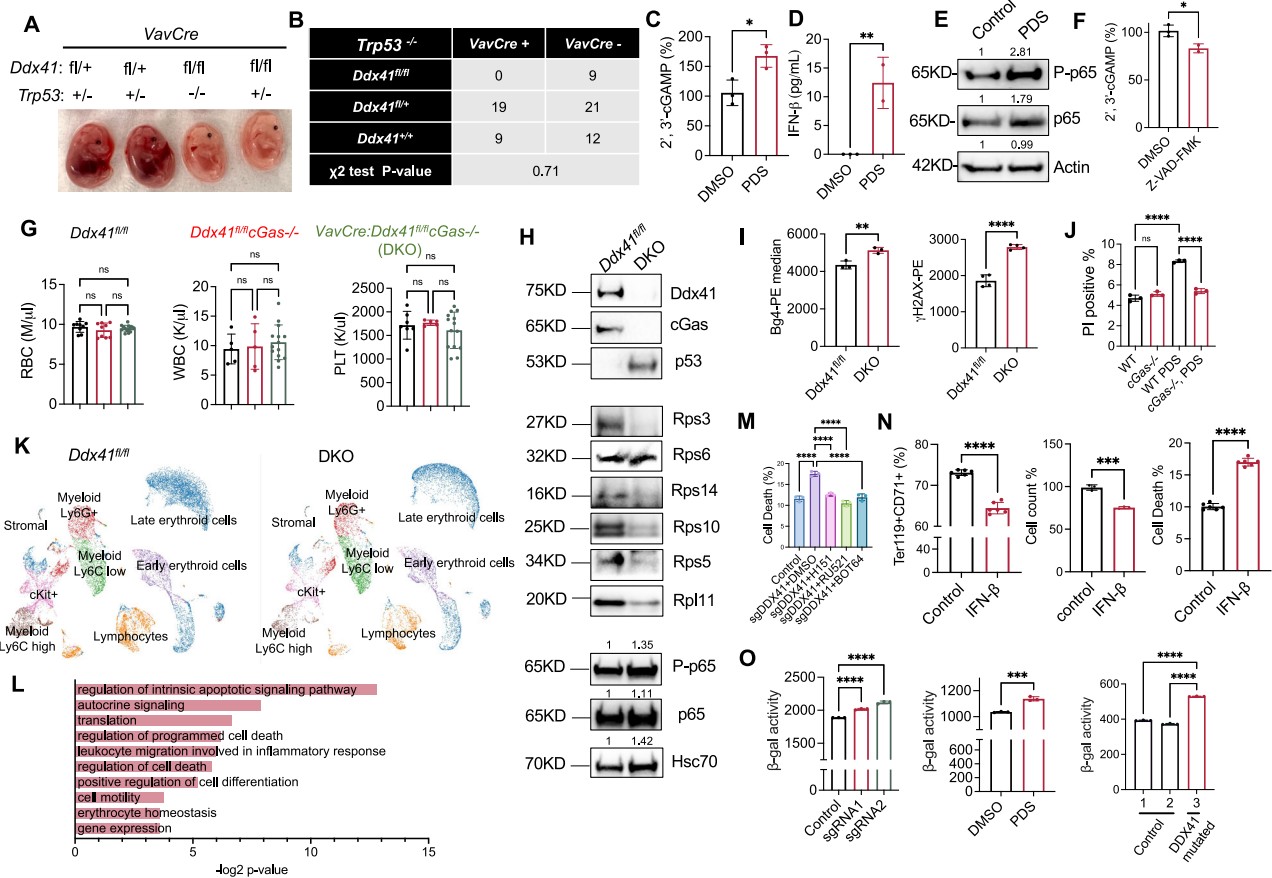

**Fig. 6 | Activation of the cGAS-STING pathway, but not p53 upregulation, is essential for DDX41 deficiency-mediated ineffective erythropoiesis. A** Pictures of E14.5 embryos in (**B**). **B** Mendelian segregation of indicated mice. A two-tailed $\chi^2$ test yields a *p* value of 0.71, indicating no significant deviation from expected Mendelian segregation. **C, D** Mouse lineage-negative HSPCs were cultured in Epo medium for 1 day, followed by treatment with 1 μM PDS for 6 h before 2′,3′-cGAMP (**C**, *p* = 0.0201) and IFN-β (**D**, *p* = 0.0066) ELISA. **E** Western blotting of indicated proteins in cells from (**C**). Data are representative of 3 independent experiments. **F** Same as (**C**) except the cells were treated with 40 μM Z-VAD-FMK for 6 hours before ELISA. *p* = 0.0385. **G** Complete blood count of indicated 2-month-old mice. **H** Western blotting of indicated proteins from bone marrow Ter119+ cells of mice from (**G**). **I** Quantification of G4 and γH2AX by flow cytometry in cells in (**H**). G4 *p* = 0.007, γH2AX *p* < 0.0001. **J** Same as (**C**) except HSPCs were from the indicated mice. Cell death was analyzed by flow cytometry. ****p* < 0.0001. **K** UMAP showing distribution of annotated bone marrow populations in indicated 2-month-old mice.

**L** Pathway enrichment analyses of down-regulated genes in late erythroid cell population in DKO mice in (**K**). **M** CD34+ cells were cultured in Epo medium for 4 days, then subjected to DDX41 CRISPR knockout and inhibitor treatment for 2 days. Cell death (Annexin V + /PI + ) was analyzed by flow cytometry. The controls received scrambled sgRNA. ****p* < 0.0001. **N** Mouse lineage-negative cells were cultured in Epo medium for 1 day with/without 100 ng/ml IFN-β, then analyzed by flow cytometry for differentiation, proliferation, and cell death (Annexin V + /PI + ). ****p* < 0.0001. **O** β-galactosidase activity was quantified in DDX41-knockout CD34+ cells, PDS-treated mouse lineage-negative cells (1 μM, 24 hours), and MDS patient bone marrow cells (Fig. 5J). All the error bars represent the SEM of the mean. The comparison between two groups was evaluated with 2-tailed t tests, and the comparison among multiple groups was evaluated with 1-way ANOVA tests. ns: not significant. * *p* < 0.05, ***p* < 0.01, ****p* < 0.001, and *****p* < 0.0001. Western blotting was quantified on top of the bands in E and H.

(Supplementary Fig. 9C). Western blotting of the bone marrow Ter119+ erythroid cells confirmed cGas and Ddx41 deletions but also showed loss of ribosomal proteins and upregulation of p53 in the DKO mice (Fig. 6H), which indicates that G4 upregulation-mediated genome instability leads to parallel activation of the cGas and p53 pathways. Consistent with this indication, the level of phospho-p65 was unchanged (Fig. 6H), whereas G4 and γ-H2AX remained upregulated in the bone marrow Ter119+ erythroid cells of the DKO mice compared to the cells in their littermate control mice (Fig. 6I). The critical role of cGas in mediating Ddx41 deficiency-induced pathogenesis was further evidenced by the resistance of primary erythroid cells from cGas knockout mice to PDS-mediated cell death (Fig. 6J). We next performed a single-cell RNA sequencing assay of the total bone marrow cells from DKO mice and their littermate control (Supplementary Fig. 9D). While non-erythroid hematopoietic populations showed no apparent distinctions, late erythroid cells in DKO mice exhibited altered gene expression, including down-regulation of genes involved

in cell death regulation and erythroid hemostasis (Fig. 6K, L and Supplementary Fig. 9E, Supplementary Data 1). Since apoptotic signals are known to be involved in chromatin condensation in terminal erythropoiesis[34,52,54,55], it is possible that these pathways are altered in the DKO cells.

cGAS was shown to translocate to the nucleus under DNA double-strand break to suppress DNA repair independent of STING[56]. We found that cGas, as well as Sting, was predominantly located in the cytoplasm in the erythroid cells upon PDS treatment, demonstrating a cGas-Sting-dependent pathway and consistent with the upregulation of IFN-β and the activation of NF-κB signaling (Supplementary Fig. 9F). In line with the role of the cGAS-STING pathway, treatment of DDX41 deficient CD34+ cells with a cGAS inhibitor (RU521), a STING inhibitor (H151), or an NF-κB inhibitor (BOT64) significantly rescued cell death (Fig. 6M). Similar to PDS, treatment of the erythroid cells with IFN-β significantly compromised cell differentiation, proliferation, and induced cell death (Fig. 6N).

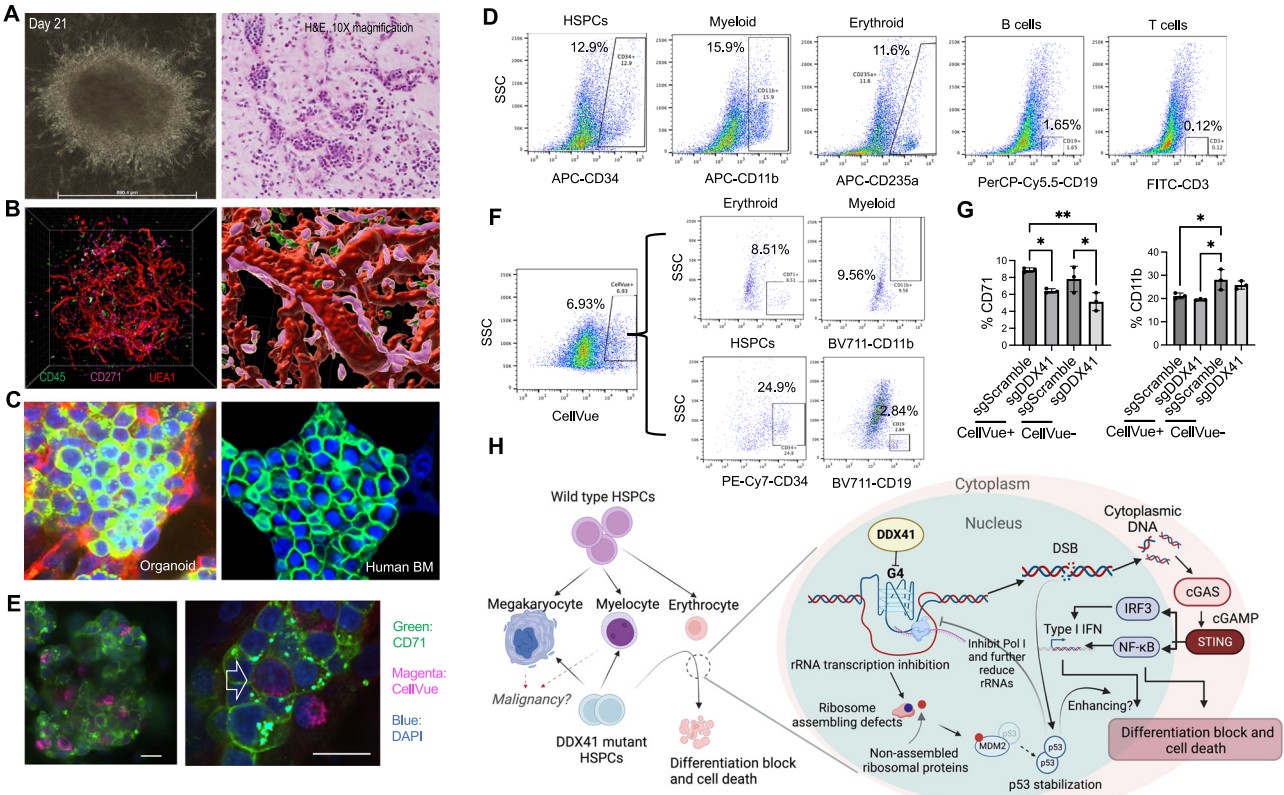

**Fig. 7 | Human erythroid cells are sensitive to DDX41 deficiency in an iPSC-derived bone marrow organoid environment. A** Representative wide-field picture and H&E stains of bone marrow organoid in culture. **B** Whole-mount 3D imaging of the organoids. Imaris was used for cell surface rendering. Organoids were stained with indicated antibodies and subsequently imaged using a laser scanning confocal platform. **C** Confocal immunofluorescence assays of erythroid islands in the iPSC-derived bone marrow organoids (left) and a primary human bone marrow biopsy (right). CD71 was labeled with green for organoids and primary bone marrow. DAPI: blue. **D** Flow cytometry assays of the organoids using indicated antibodies for various lineages. **E** 10,000 CellVue-labeled donor CD34+ HSPCs were co-incubated with iPSC-derived bone marrow organoids for 3 days in each well of a 96-well plate, followed by an immunofluorescence assay. Representative pictures show the engraftment of donor hematopoietic cells into the organoid. The arrow points to an engrafted CellVue positive cell expressing CD71. Scale bar: 10 μm. **F** Flow cytometry of the organoids using indicated antibodies for various lineages

of the engrafted cells in organoids from (**E**). **G** Same as (**E**) except the donor CD34+ cells were transduced with lentiviral vectors expressing Cas9 and indicated sgRNAs before co-incubation. After 3 days, the cells were collected for flow cytometric assays of erythroid and myeloid differentiation of CellVue-positive donor hematopoietic cells and negative iPSC-derived hematopoietic cells. Each data point represents cells combined from 10 organoids. The comparison was evaluated with 1-way ANOVA tests. CD71: sgScramble vs. sgDDX41 in CellVue$^+$ ($p = 0.0491$) and CellVue$^-$ ($p = 0.0333$); sgScramble CellVue$^+$ vs. sgScramble CellVue$^-$ ($p = 0.0054$). CD11b: sgScramble CellVue$^+$ vs. sgScramble CellVue$^-$ ($p = 0.0341$); sgDDX41 CellVue$^+$ vs. sgScramble CellVue$^-$ ($p = 0.0119$). At least 3 independent biological replicates were tested for each experiment. *$p < 0.05$, **$p < 0.01$. For representative data, at least 3 independent experiments were performed to ensure reproducibility. **H** Schematic model of the function of DDX41 during erythropoiesis. Created in BioRender. Ji, P. (https://BioRender.com/96esfmz).

These data point towards cGAS-STING activation-induced senescence, instead of p53-induced cell death, in mediating ineffective erythropoiesis in DDX41 deficiency. Consistently, we observed increased senescence-associated β-galactosidase activities in DDX41 CRISPR knockout or PDS-treated CD34+ cells in Epo-containing medium, as well as bone marrow mononuclear cells from the *DDX41* mutated patient (Fig. 6O).

### Human erythroid cells are sensitive to DDX41 deficiency in an iPSC-derived bone marrow organoid environment

To extend these findings to the human bone marrow in vivo setting, we applied an induced pluripotent stem cell (iPSC)-derived human bone marrow organoid system resembling primary human bone marrow biopsy samples[57] (Fig. 7A). Whole-mount 3D imaging revealed an endothelial network accompanied by stromal cells along the capillary wall and hematopoietic cells arranged in clusters within and outside the vessels (Fig. 7B). Erythropoiesis in the organoid mirrors primary human bone marrow samples, forming tight erythroid islands (Fig. 7C). A flow cytometry assay further confirmed multilineage hematopoiesis

in the organoids (Fig. 7D). To determine whether these organoids can be applied to study human ex vivo bone marrow engraftment, we incubated the organoids with CellVue-labeled donor bone marrow CD34+ cells for three days. The donor cells could be readily detected and surrounded by the recipient hematopoietic cells in the organoids (Fig. 7E). These donor cells could also differentiate into mature hematopoietic cells detected by immunofluorescence and flow cytometry assays (Fig. 7E, F). With this system, we depleted DDX41 through CRISPR in CD34+ cells and incubated these cells with the bone marrow organoids. As expected, the CD71+ erythroid cells, but not CD11b+ myeloid cells, derived from the donor CD34+ cells transduced with DDX41 sgRNA were significantly reduced compared to the control donor cells. Interestingly, the erythroid cells derived from the recipient bone marrow organoids were also significantly reduced, indicating a potential indirect impact from the inflammatory environment due to the cGAS-STING activation (Fig. 7G).

We also treated the organoid with PDS for 24 hours and analyzed erythropoiesis using flow cytometry. Since the organoid already contains erythroid cells at various stages of development, this approach

allows us to study the responses of cells at different stages of erythropoiesis to G4 accumulation. Consistent with the results from other experiments, we found a significant decrease in the population of CD34 + CD71+ cells, which include mainly early-stage erythroid cells. However, no significant differences were found in the population of CD34-, CD71 + , and CD235a+ cells, which represent late-stage erythroid cells (Supplementary Fig. 9G).

## Discussion

Our study shows that DDX41 directly interacts with and resolves G-quadruplexes. DDX41 deficiency increases G4 levels, causing genome instability and impairing ribosomal biogenesis. This elevation in G4s activates p53 and the cGAS-STING pathway. Genetic studies in mice confirm that knocking out cGas, but not p53, rescues the lethality in hematopoietic-specific Ddx41 knockout mice (Fig. 7H). A previous study demonstrated that DDX41 possesses both DNA unwinding and annealing activities, and that the R525H mutant selectively loses its unwinding ability without affecting annealing. This selective functional loss was associated with increased type I interferon production[41]. Our findings are consistent with this report, as we also observed impaired G4 unwinding activity of the R525H mutant in our FRET-based assays, further supporting the importance of DDX41's resolvase function in maintaining genome stability and regulating innate immune activation.

In $VavCre:Ddx41^{fl/fl} cGas-/-$ (DKO) mice, the sustained reduction in ribosomal proteins and increased p53 levels mirror mechanisms seen in ribosomopathies like Diamond-Blackfan Anemia (DBA). In DBA, impaired ribosome assembly results in excess free ribosomal proteins, which bind to MDM2, reducing its ability to degrade p53[49]. While DBA is classically associated with erythroid failure, recent studies have identified ribosomal dysfunction and p53 activation in HSPCs and myeloid lineages as well[13,58]. These findings indicate that ribosomopathies can affect multiple hematopoietic compartments beyond erythropoiesis, particularly under stress or during aging. Despite ribosomopathies, the complete rescue of survival in the DKO mice is consistent with reports that an increase in p53 activity or haploinsufficiency of $MDM2$ does not necessarily result in severe phenotypes in mice[50,59–62]. Nevertheless, ribosomopathies and the continued presence of genome instability due to G4 upregulation increase the possibility of the risk of the development of hematologic malignancies in these mice, especially with aging. It is also possible that the phenotypically normal myeloid, monocytic, and dendritic-specific Ddx41 knockout mice, which is consistent with a previously published report[41], could develop hematologic diseases with aging due to the accumulation of DNA double-strand break with the upregulation of G4. The increased G4, γ-H2AX, and p53 levels in the MN patient with somatic DDX41 mutation are consistent with this possibility.

Our study reveals erythroid cells' hypersensitivity to DDX41 deficiency, likely due to widespread G4 structure enrichment, especially on ribosomal DNAs. Erythroid cells are vulnerable to replication stress from DNA secondary structures like G4, explaining the detrimental effects of DDX41 deficiency or PDS treatment on their differentiation and survival. Similar sensitivity of erythroid precursors to Ddx41 deficiency was also reported in zebrafish[29], highlighting the essential roles of DDX41 in governing erythroid genome stability across species. Under physiologic conditions, ribosome biogenesis is critical in the early stages of erythropoiesis but decreases in the late stages of terminal erythropoiesis by the decline in rDNA transcription[63]. This is in line with our findings that there are no significant phenotypes in $HBBCre:Ddx41^{fl/fl}$ mice despite an increase in G4 since G4 upregulation-mediated defects occur in the late stages of terminal erythropoiesis when the ribosomal biogenesis is naturally reduced, and chromatin is markedly condensed[34,52,64]. It is also possible that the activation of the cGAS-STING pathway due to DDX41 deficiency is not essential at the late stages of terminal erythropoiesis.

These findings in murine models were confirmed in human settings using primary patient samples and iPSC-derived bone marrow organoids. The somatic G530D mutation identified in our patient is notably underrepresented in the erythroid population, aligning with the effects of DDX41 loss-of-function on erythropoiesis, particularly in the aging context. Similar impairments were observed when DDX41-deficient HSPCs were engrafted into iPSC-derived bone marrow organoids, leading to defective erythroid maturation and increased cellular senescence. These effects were also evident in organoid-derived erythroid cells, consistent with activation of the cGAS-STING pathway in an inflammatory bone marrow microenvironment, which may heighten erythroid cell sensitivity[65]. However, further investigation across additional patient samples and DDX41 mutation types will be necessary to determine whether this erythroid-specific vulnerability is a generalizable consequence of DDX41 deficiency or specific to certain pathogenic variants.

## Methods

### Mice

Wild-type C57BL/6 J mice (strain #000664), cGas knockout mice (strain #026554), Cre-recombinase expression mice $VavCre$ (strain #008610), $EpoR$-$Cre$ (strain #035702), $MRP8$-$Cre$ (strain #021614), $CD11c$-$Cre$ (strain #008068), and $LysM$-$Cre$ (strain #004781) were purchased from the Jackson Laboratory. $HBB$-$Cre$ was provided by Nicolae Valentin David. $Ddx41$ floxed ($Ddx41^{fl/fl}$) mice were provided by Susan Ross.

Tissue-specific Ddx41 knockout mice were generated by breeding tissue-specific Cre recombinase expression mice with $Ddx41^{fl/fl}$ mice. The F1 generation, which carried Cre recombinase and was $Ddx41^{fl/+}$, was subsequently crossed with $Ddx41^{fl/fl}$ mice. For $Ddx41^{fl/fl}$ breeding with $Vav$-$Cre$, female $Vav$-$Cre$-carrying mice were bred with male $Ddx41^{fl/fl}$ mice to avoid $Ddx41$ knockout during spermatogenesis.

To generate $VavCre:Ddx41^{fl/fl},cGas^{-/-}$ double knockout (DKO) mice, female $VavCre$ mice were first crossed with $Ddx41^{fl/fl}$ mice to produce $VavCre:Ddx41^{fl/+}$ offspring. These mice were then crossed with $cGas^{-/-}$ mice to obtain $VavCre:Ddx41^{fl/+},cGas^{+/-}$ mice. Genotyping was performed using genomic DNA extracted from tail biopsies. In the subsequent generation, female $VavCre:Ddx41^{fl/+},cGas^{+/-}$ mice were crossed with male $Ddx41^{fl/+},cGas^{+/-}$ mice to generate DKO mice ($VavCre:Ddx41^{fl/fl},cGas^{-/-}$). Littermates carrying $Ddx41^{fl/fl}$ but lacking $VavCre$ and with wild-type $cGas$ alleles were used as controls. Whenever feasible, especially in embryonic and early postnatal studies, we used littermate controls. This approach ensured that comparisons between genotypes were made within the same genetic background and developmental environment. All animal studies followed Northwestern University's Institutional Animal Care and Use Committee guidelines.

### Mouse bone marrow and spleen flow cytometry assay

The bone marrow and spleen flow cytometry assays were performed as previously reported[21,65–67]. Briefly, mice were euthanized following ethical guidelines approved by the IACUC at Northwestern University. Femur bones were dissected to extract bone marrow cells via PBS flushing. Spleens were weighed and subsequently homogenized. These isolated single cells were labeled with antibodies for subsequent flow cytometric analysis. For erythroid cell characterization, APC-Ter119, PE-CD71, and FITC-CD44 antibodies were employed. Myeloid cell analysis utilized APC-CD11b, BV421-ly6C, and PE-ly6G antibodies, while lymphoid cell populations were assessed using PE-B220 and APC-CD3 antibodies. For HSPC analysis, the cells were stained with BD Pharmingen™ PerCP-Cy™5.5 Mouse Lineage Antibody Cocktail along with FITC-Ly5A/E and APC-CD117. Subsequently, cells that exhibited a negative signal in the PerCP-Cy5.5 channel were selected, and further analysis was conducted on these cells using the FITC and APC channels to identify and characterize LSK (Lin-Sca1+Kit + ) cells. For cell death

analysis, cells were stained using the BD Pharmingen™ FITC Annexin V Apoptosis Detection Kit I and subsequently analyzed via flow cytometry.

## Quantification of G-quadruplexes using flow cytometry

The quantification of G-quadruplexes was performed using the BG4 antibody with mouse IgG1 Isotype. Cells were fixed in 0.25% Glutaraldehyde/PBS for 1 hour, followed by permeabilization with 0.1% Triton-X/PBS. To remove the RNAs, cells were incubated in 0.05% Triton-X/PBS with 3% bovine serum albumin (BSA) and 10 μg/ml RNaseA for 30 min. Subsequently, they were incubated in PBS containing 0.05% Triton-X, 3% BSA, and 10 μg/ml BG4 antibody for at least 30 min. After washing, cells were resuspended in PBS with PE-conjugated anti-mouse IgG1 antibody for 10 min. Finally, the cells were washed and resuspended in PBS for flow cytometry analyses. Median fluorescence intensity was used for statistical analyses to mitigate the influence of outliers and skewed distributions.

## Antibodies

The following antibodies (vendor, catalog number) were used for Western blotting assays: DDX41 (CST, #15076), BG4 (Absolute Antibody, Ab00174), p53 (CST, #9282), cGAS (E5V3W) (CST, #79978), Phospho-Histone H2A.X (Ser139) (D7T2V) (CST, #80312), NF-κB Pathway Antibody Sampler Kit (CST #9936), RPL26 (Proteintech, #17619-1-AP), RPL7A (Proteintech, 15340-1-AP), RPS27A (Proteintech, #14946-1-AP), RPS3 (Proteintech, #11990-1-AP), RPS6 (Proteintech, #14823-1-AP), RPS19 (Proteintech, #15085-1-AP), RPS14 (Proteintech, 16683-1-AP), Ribosomal Protein S3 (D50G7) (CST, #9538S), S6 Ribosomal Protein (5G10) (CST, #2217), RPL10 (Proteintech, #72912), RPL11 (D1P5N) (Proteintech, #18163), RPL5 (Proteintech, #14568), HRP-conjugated beta actin monoclonal antibody (Proteintech, #HRP-66009), anti-rabbit IgG, HRP-linked antibody (CST, #7074), and anti-mouse IgG, HRP-linked antibody (CST, #7076). Western blotting antibodies were diluted to a final concentration of 1:1000.

For flow cytometry assays, the following antibodies were used: APC rat anti-mouse TER119 (BD, #561033), FITC anti-mouse CD71 (Biolegend #113805), PE rat anti-mouse CD71 (BD, #567206), PE anti-mouse IgG1 (Biolegend, #406607), PE rat anti-mouse CD45R/B220 (BD, #553089), APC rat anti-mouse CD3 (BD, #565643), Phospho-histone H2A.X (Ser139) (CR55T33), PE F(ab')2 fragment (Alexa Fluor® 488 conjugate) (CST, #4412), anti-mouse IgG (H + L), F(ab')2 fragment (Alexa Fluor® 647 conjugate) (CST, #4410), Phospho-Histone H2A.X (Ser139) monoclonal antibody (Invitrogen, 12-9865-43), anti-rabbit IgG F(ab')2 fragment (Alexa Fluor® 488 Conjugate) (CST, #4412), anti-mouse IgG (H + L) F(ab')2 fragment (Alexa Fluor® 647 Conjugate) (CST, #4410), FITC rat anti-mouse CD44 (BD, #561859), APC rat anti-CD11b (BD, #553312), BV421 rat anti-mouse Ly-6C (BD, #562727), PE rat anti-mouse Ly-6G (BD, #561104), FITC rat anti-mouse Ly-6A/E (BD, #557405), and APC rat anti-mouse CD117 (BD, #553356). Flow cytometry antibodies were diluted to a final concentration of 1:100.

## Isolation of bone marrow and fetal liver hematopoietic stem and progenitor cells (HSPCs)

Mouse femurs were dissected with surrounding tissues removed. The bone marrow was collected by flushing the bone cavities using 10 ml of phosphate buffer saline (PBS) supplemented with 2% fetal bovine serum (FBS). The harvested cells were then incubated for 10 min with a cocktail of biotin-conjugated lineage antibodies against CD3e, CD11b, CD45R, Ly-6G/C, and Ter119 (BD, #559971), followed by washing with PBS containing 2% FBS. Subsequently, the cells were incubated with streptavidin particles plus beads (BD, #557812) for 10 min and placed on a magnetic rack for 15 min for magnetic separation. The supernatant, enriched with lineage negative HSPCs, was collected for subsequent cell counting and analyses.

For fetal liver HSPC purification, E13.5 fetal liver was homogenized by repeated pipetting and filtering through a 40 μm cell strainer to obtain single cells. The filtered cells were incubated with biotin anti-mouse Ter119 antibody for 15 min and washed with PBS. After washing, the cells were resuspended, and Ter119-positive cells were pulled down with streptavidin magnetic beads. The remaining Ter119 negative HSPCs were then used for the follow-up experiments.

## In vitro erythroid differentiation of mouse lineage negative cells

The isolated lineage-negative cells were cultured in erythropoietin (Epo)-containing medium (IMDM supplemented with 15% FBS, 10% BSA, 10 μg/ml insulin, 200 μg/ml holo-transferrin, 0.1 mM β-mercaptoethanol, 1% penicillin-streptomycin, 2 mM glutamine, and 2 unit/ml Epo) at a density of $5 \times 10^5$ cells/ml. The cells were then incubated for up to 48 hours at 37 °C with 5% $CO_2$. Cells were then harvested for analysis at different time points based on the experiments.

## Human cell line, bone marrow cells, and Sanger sequencing

HEK293T cells were purchased from ATCC (CRL-1573). Total bone marrow cells from MDS patients were obtained following informed consent under institutional review board-approved protocols at Northwestern University. Isolation of specific cell populations was achieved using a bead-based method through biotin-conjugated CD34+ and CD71+ antibodies. Following isolation, the cells underwent a washing process repeated three times in preparation for DNA extraction. The extracted DNA was subsequently used in PCR reactions for further analyses of mutated genes in the patients. For the *DDX41* gene, the following primers were used: Forward - ATGGGTTAGGCCGGAAAAGGG, Reverse - TGACTCATCTGGGGGAGGAG. For the *PRPF8* gene, the primers used were Forward – AAGGAGACAATCCCCCGA and Reverse - TATAGGCCAGGTCAATGGCG.

## Flow cytometric analysis of mouse in vitro erythroid differentiation

Flow cytometric analysis of differentiation and enucleation of cultured mouse erythroblasts was conducted following previously established methods[21]. In brief, surface antigen labeling was performed using FITC or PE conjugated antibodies against CD71 and Ter119, along with their respective isotype controls. The stained cells were analyzed using a FACSCanto flow cytometer from Becton Dickinson. Additionally, propidium iodide was included to exclude non-viable cells from the analysis.

## G-quadruplexes confocal immunofluorescence staining

Cells were washed with ice-cold serum-free Iscove's Modified Dulbecco's Medium (IMDM) and plated on poly-L-lysine-coated coverslips (BD Biosciences), followed by a 5-min incubation at 37 °C in a humidified incubator. After attachment, cells on the coverslip were washed with ice-cold PBS (pH 7.2), fixed in 4% paraformaldehyde for 15 min, and permeabilized with 0.1% Triton-X 100 in PBS for 10 min at room temperature. Following three PBS washes, cells were blocked with 3% BSA in PBS containing 0.05% Triton-X 100 for 1 hour at room temperature. Subsequently, the cells were stained with FLAG-BG4 antibody (1:100 final concentration) and relevant secondary antibody (1:100 final concentration) for 1 hour each, followed by three cycles of PBS washing for 5 min each. Finally, the cells were mounted on glass slides using ProLong™ Diamond Antifade Mountant with DAPI (Invitrogen).

Three-dimensional (3D) imaging of Ddx41 and G4 immunostaining was performed using a Nikon AXR laser scanning confocal microscope. A Nikon CFI Apo TIRF 60x Oil immersion objective (NA 1.49) was used for all Z-stack image acquisitions. Scanning was carried out using the Galvo scanning mode with a dwell time of 4 seconds per pixel. A total of 50 Z-slices were collected over a 25 μm depth, resulting

in a Z-step interval of 0.5 μm. 3D reconstruction of the acquired image stacks was performed using Imaris software (version 10.0).

## Human CD34+ cell culture and differentiation

Human CD34+ cells (STEMCELL Technologies) were cultured in IMDM supplemented with 3% serum, 2% plasma, 10 μg/ml insulin, 3 IU/ml heparin, 200 μg/ml transferrin, 3 U/ml Epo, 10 ng/ml stem cell factor (SCF), 1 ng/ml interleukin-3 (IL-3), and 1% penicillin/streptomycin. The differentiation protocol involved a sequential withdrawal of cytokines. Initially, from day 0 to day 7, the culture included IL-3, EPO, and SCF, with medium exchanges every 2 days to maintain a cell concentration of approximately $1 \times 10^5$ cells/ml. On day 7, a complete medium change was performed to eliminate IL-3. On day 10, SCF was withdrawn after a complete medium change. The cells were then maintained at $1 \times 10^6$ cells/ml through regular medium changes until day 14. Starting from day 15, Epo was withdrawn through a complete medium change, and the cells were maintained at a concentration of $5 \times 10^6$ cells/ml with regular medium changes until day 20. Cells were harvested at specific time points for analysis.

## CRISPR/Cas9 mediated DDX41 knockout in human CD34+ cells

The sgRNAs targeting DDX41 or scrambled sgRNA were cloned into the lentiviral vector lentiCRISPR v2 (Addgene, #52961, encoding Cas9) using the previously reported protocol[68]. The lentiviruses were produced in HEK293T cells following the manufacturer's protocol (Invitrogen, MA, USA). For viral infection, roughly $3 \times 10^7$ control or DDX41 sgRNA lentiviral particles were used to infect $5 \times 10^5$ CD34+ cells. 14 hours after infection, cells were washed with PBS and cultured in a fresh medium. Cells were cultured for an additional 6 hours before follow-up studies.

## Cleavage under targets and release using nuclease (CUT&RUN) assay

The following populations were subjected to CUT&RUN assays. HSPCs were isolated utilizing a negative selection methodology via the BD Biotin Mouse Lineage Panel kit (BD #559971). For the erythroid cells, biotin-conjugated Ter119+ antibody and magnetic streptavidin beads were used. The CUT&RUN assay was carried out using the Cell Signaling Technology CUT&RUN Assay Kit (CST #86652) following the manufacturer's protocol with minor adaptations. Cells ($1 – 5 \times 10^5$) were harvested, washed, and bound to Concanavalin A-coated magnetic beads. Bead-bound cells were incubated in digitonin-containing antibody buffer with primary antibodies against G-quadruplex structures (BG4, Absolute Antibody ab00174-1.1) and DDX41 (CST #15076) for 2 hours at 4 °C. After washing, pAG-MNase was added and allowed to bind for 1 hour at 4 °C. Targeted digestion was initiated by adding calcium chloride and incubating for 30 min at 4 °C. The reaction was stopped using Stop Buffer, and cleaved chromatin fragments were allowed to diffuse into the supernatant under continued digitonin permeabilization. DNA was purified and quantified from the supernatant using spin columns. Libraries were prepared and sequenced on the Illumina NextSeq 500 platform at the Northwestern University NUSeq Core Facility.

Initial demultiplexing, adapter trimming, and differential peak analysis were performed by the NUSeq Core. Subsequent bioinformatic analyses were conducted on the Northwestern Quest high-performance computing cluster. Sequencing reads were aligned to the mm10 mouse reference genome using Bowtie2 with parameters optimized for short fragments. Signal quantification and heatmap generation were performed using the computeMatrix and plotHeatmap functions in deepTools. Genome-wide coverage tracks were visualized using the UCSC Genome Browser and IGV with BigWig files provided by NUseq. To define consensus peaks within each condition, bedtools merge was used to combine replicate peak sets. Codistribution and unique peak analysis across experimental groups were performed

using bedtools intersect. The number of shared and condition-specific peaks was quantified and exported as CSV files. These results were visualized as Venn diagrams using the Venn Diagram and ggplot2 packages in R. For functional annotation and visualization of genomic distributions, merged peak files were further analyzed using the ChIPseeker R package. A TxDb object was constructed from the GTF file (mm39.knownGene.gtf.gz) using GenomicFeatures::makeTxDbFromGFF. Peak annotations were generated using annotatePeak, with the TSS region defined as ±3 kb. Gene-based annotations were performed using the org.Mm.eg.db database. Pie charts of genomic region distributions were generated using plotAnnoPie. Motif enrichment analysis was performed using the HOMER tool findMotifsGenome.pl. Input peak regions were analyzed against the mm10 mouse genome using the -size given option to preserve peak widths. The sequencing data are available from the Gene Expression Omnibus database under accession code GSE254143.

## G4 pull-down assay

The oligonucleotides designed for G4 pull-down assay included sequences capable of forming G4 structures (G4-1: GATACTGCGAGTAGATACATGAAGGGAGGGCGCTGGGAGGAGGGGATCT (derived from Kit1), G4-2: GATACTGCGAGTAGATACATGAACGGGCGGGCGCGAGGGAGGGGATCT (derived from Kit2), G4-3: GATACTGCGAGTAGATACATGAAGGCGAGGAGGGGCGTGGCCGGCATCT (derived from Spb1) and non-G4 controls (NC-1: GATACTGCGAGTAGATACATGAAGCAAGCACGCTGCAAGGAGCAATCT, NC-2: GATACTGCGAGTAGATACATGAACGCACGCACGCGAGCAAGCAGATCT, NC-3: GATACTGCGAGTAGATACATGAAGGCGACAAGGCACGTGGCCCACATCT), along with a complementary ssDNA sequence ATGTATCTACTCGCAGTATCATAC tagged with a 3' biotin (BiosG). Oligonucleotides were initially dissolved to 100 μM in a buffer containing 10 mM Tris-HCl (pH 8.0) and 1 mM EDTA (TE buffer). For experimental use, they were further diluted to a 10 μM concentration. The ssDNA was mixed with either G4 or control oligonucleotides in a solution comprising 20 mM HEPES (pH 7.5), 250 mM KCl, and 1 mM DTT. This mixture was then heated at 95 °C for 10 min, followed by a gradual cooling to 25 °C at a rate of 0.1 °C per second using a PCR machine.

For binding assays, high-affinity streptavidin magnetic beads (Pierce, Cat. No. 88817) were prepared in a binding buffer (10 mM Tris-HCl pH 8, 100 mM KCl, 0.1 mM EDTA, 1 mM DTT, 0.05% Tween-20) and incubated with the folded G4 oligonucleotides (0.5 nM oligonucleotides in 10 μl of streptavidin magnetic beads) for 2 hours at 4 °C. Following two washes in the binding buffer to remove unbound oligonucleotides, the beads were incubated with 5 μg of DDX41 proteins for 1 hour at 4 °C. Subsequent washes with binding buffers containing increasing concentrations of KCl (200–500 mM) were performed to remove non-specifically bound proteins. Proteins that remained bound to the beads were eluted by incubating for 5 min at 95 °C in 2x SDS-PAGE loading buffer. The eluted proteins were then analyzed by SDS-PAGE and Western blotting to assess the binding interactions.

## G4 fluorescence resonance energy transfer (FRET) assay

G4 oligonucleotides were designed following previously published G4 oligo sequences[69,70]. Three oligonucleotides derived from the human MYC gene were generated. Each was conjugated with a FAM signal at the 5'-end and a BHQ group at the 3'-end. The sequences of these oligonucleotides are as follows: G4-1: 5'-FAM-TGGGGAGGGTGGGGAGGGTGGGGAAGGT-3'-BHQ, G4-2: 5'-FAM-GGGTGGGTTGGGTGGGG-3'-BHQ, and G4-3: 5'-FAM-GGGAGGGTTGGGTGGGG-3'-BHQ. To form G4 structures, these oligonucleotides were re-suspended in water and annealed in a tris-acetate solution containing 100 mM K+, with a total oligonucleotide concentration of 10 μM. The annealing was achieved by slow cooling from 95 °C to room temperature. The resulting annealed G4 substrates were aliquoted into small tubes and stored at

−20 °C. Unfolding assays were conducted using a buffer composed of 20 nM oligonucleotides, 50 mM tris-acetate, 2.5 mM MgCl2, 0.5 mM DTT, and 50 mM KCl, along with varying concentrations of recombinant DDX41 protein. Fluorescence changes were monitored using an excitation wavelength of 490 nm and emission at 520 nm. The percentage of unfolding was calculated as: Unfolded G4 Percentage = (Sample G4 FAM Signal − Folded G4 FAM Signal) / (Unfolded G4 FAM Signal − Folded G4 FAM Signal).

### DDX41 protein expression and purification

Halo-tagged human DDX41 protein was expressed using the pJFT7-nHalo-DC vector containing the human DDX41 cDNA sequence (DNASU HsCD00947316). BL21 bacteria carrying the DDX41 plasmid were resuspended in LB medium supplemented with 50 μg/mL Ampicillin. The suspension was incubated at 37 °C to OD value around 0.6. Subsequently, IPTG was added to the bacterial culture to a final concentration of 1 mM and incubated at room temperature for 1 day. The cells were then collected for protein purification using Promega HaloTag Protein Purification System. Briefly, the collected cells were first resuspended and lysed using lysozyme and sonicated in the presence of protease inhibitors to release the target protein. The targeted proteins were then bound to the resin. Subsequently, the bound protein was cleaved using TEV protease to release DDX41, while the Halo tag was removed. The purified proteins were then stored at −80 °C for future use.

For the generation of DDX41 mutants (R525H and G530D), the DDX41 expression vector was modified using the GeneArt Site-Directed Mutagenesis System to introduce single-nucleotide mutations into the vector. The mutagenesis was carried out using the following primers: G530D Forward: GCTCGGGAAACACAGACATCGCCAC TACCTT, Reverse: AAGGTAGTGGCGATGTCTGTGTTTCCCGAGC, R525H Forward: TTGGCCGCACCGGGCACTCGGGAAACACAGG, Reverse: CCTGTGTTTCCCGAGTGCCCGGTGCGGCCAA. Protein expression and purification of the mutant DDX41 variants followed the same method described for the wild-type DDX41.

### Fluorescence polarization assay

Fluorescence polarization (FP) assays were performed to assess the interaction between nucleic acid substrates and the purified DDX41 protein. To generate double-stranded DNA (dsDNA), double-stranded RNA (dsRNA), and RNA–DNA hybrid 12-mer substrates, 6-FAM-labeled and unlabeled oligonucleotide pairs were heated to 95 °C and gradually cooled to 4 °C. The resulting single-stranded and double-stranded substrates were diluted to a final concentration of 20 nM in FP assay buffer containing 20 mM HEPES (pH 7.0), 100 mM NaCl, and 5% glycerol.

To generate G-quadruplex (G4) structures, oligonucleotides were resuspended in water and annealed in a Tris-acetate solution containing 100 mM K$^+$ at a total oligonucleotide concentration of 10 μM. Annealing was achieved by slow cooling from 95 °C to room temperature. The G4 oligonucleotides were then diluted into a buffer containing 20 nM oligonucleotides, 50 mM Tris-acetate, 2.5 mM MgCl$_2$, 0.5 mM DTT, and 50 mM KCl. Unfolded G4 oligonucleotides were prepared using the same annealing process but without K$^+$.

Purified full-length DDX41 protein was added to individual substrates at the indicated final concentrations. Fluorescence polarization of the 6-FAM-labeled probes was measured using a Tecan Infinite M1000 Multimode Microplate Reader at 20 °C at the Northwestern University High Throughput Analysis Lab, with an excitation wavelength of 470 nm, an emission wavelength of 515 nm, a gain of 120, and 20 flashes per measurement. The fluorescence polarization value (P) was calculated using the equation: $FP = (F\| - F\perp) / (F\| + F\perp)$, where $F\|$ is the fluorescence intensity measured with the emission polarizer parallel to the excitation polarizer. $F\perp$ is the fluorescence intensity measured with the emission polarizer perpendicular to the excitation

polarizer. Relative fluorescence polarization values were expressed as a percentage by subtracting the FP value of the oligonucleotide-only control and normalizing to the control value. The following nucleic acid sequences were used. DNA-RNA hybrids were the RNA-12mer hybrid with a DNA-12mer. G4-1: GGGTGGGTTGGGTGGGG-6-FAM, G4-2: GGGAGGGTTGGGTGGGG-6-FAM, G4-3: GGGTGGGATGGGTGGGG-6-FAM, DNA-12mer-fw GACACCTGATTC-6-FAM, DNA-12mer-rev GAATCAGGTGTC, RNA-12mer-fw GACACCTGATTC-6-FAM, RNA-12mer-rev GAATCAGGTGTC.

### Quantitative RT-PCR of mouse 47 s and 45 s ribosome RNA

Total RNA was extracted from lysed cells using the Qiagen RNA extraction kit. Equivalent amounts of RNAs were employed for reverse transcription reactions with Takara PrimeScript RT Master Mix to generate cDNAs. The cDNAs were subsequently utilized as input for PCR reactions, performed using Sybr Green PCR mix. For the amplification of 45 s rRNA, the following primer pairs were used: rn45sF1 (GCTGCGTGTCAGACGTTTTT) with rn45sR1 (AGAAAAGAGCGG AGGTTCGG), rn45sF2 (AGAGAACCTTCCTGTTGCCG) with rn45sR2 (AACTTTCTCACTGAGGGCGG), rn45sF3 (ATCGACACTTCGAACGCA CT) with rn45sR3 (CACACGTCTGAACTTCGGGA), and rn45sF4 (TCC TTGTGGATGTGTGAGGC) with rn45sR4 (GGGAACATGGTCAAGC GAGA). For 47 s rRNA amplification, the primer pairs included rn47sF1 (GGTGTCCAAGTGTTCATG) with rn47sR1 (CAAGCGAGATAGGAATG TCTTAC), rn47sF2 (AGAGAACCTTCCTGTTGCCG) with rn47sR2 (AACTTTCTCACTGAGGGCGG), and rn47sF3 (TCCTTGTGGATGTG TGAGGC) with rn47sR3 (GGGAACATGGTCAAGCGAGA).

### Chronic in vivo pyridostatin treatment in mice

To ensure prolonged exposure to pyridostatin (PDS), an ALZET osmotic pump (model 2006) was used. The pump allows for continuous drug delivery for up to 6 weeks. PDS was obtained from Millipore Sigma (Catalog No. SML2690), dissolved in saline, and loaded into the pumps. A final drug dosage of 3.5 mg/kg/day was achieved through this approach. As a control, saline was also loaded into separate pumps. These pumps were then subcutaneously implanted in wild-type mice. Following an 8-week period, we conducted analyses on the peripheral blood and bone marrow of the mice to evaluate the effects of chronic G4 accumulation stress.

### Single cell RNA sequencing

Mouse femoral bone marrow was extracted and immediately submitted for single-cell RNA sequencing (scRNA-seq) analysis. Cell viability assessments were conducted to ensure a minimum of 80% viability. The extracted samples were sent to the NUseq facility, where scRNA-seq experiments were conducted. Following scRNA-seq, the resultant raw sequence data in FASTQ format were processed using Cellranger on Northwestern University Quest High-Performance Computing Cluster. Subsequently, the processed scRNA-seq data were visualized and analyzed using the Cloupe browser. The sequencing data are available from the Gene Expression Omnibus database under accession code GSE254144.

### Induced pluripotent stem cell (iPSC)-derived human bone marrow organoid

The human bone marrow organoid differentiation was previously described[57,71]. In brief, iPSCs were purchased from StemCell Technologies (SCTi003-A). During differentiation, iPSCs were dissociated using EDTA when colonies reached approximately 100 μm in diameter. The iPSC aggregates were incubated overnight in mTeSR Plus medium (StemCell Technologies) enhanced with RevitaCell in 6-well Costar Ultra-Low Attachment plates (Corning, Cat#3471). Following this incubation, cells were gathered by gravitation in a 15 mL Falcon tube (Fisher Scientific, Cat#11507411) and resuspended in Phase I medium. This medium consisted of APEL2 (StemCell Technologies, Cat#05275)

enriched with Bone Morphogenic Protein-4 (BMP4, Thermo Fisher Scientific, Cat#PHC9531), Fibroblast Growth Factor-2 (FGF2, StemCell Technologies, Cat#78134.1), and Vascular Endothelial Growth Factor-A (VEGF-165, StemCell Technologies, Cat#78159.1) at 50 ng/mL. The cells were then plated in 6-well Ultra-Low Attachment plates and incubated at 5% O2 for three days (days 0-3).

After this period, cell aggregates were collected by gravitation and suspended in Phase II medium for an additional 48 hours (days 3-5). The Phase II medium included APEL2 supplemented with BMP-4, FGF2, and VEGFA at 50 ng/mL, along with human Stem Cell Factor (hSCF, StemCell Technologies, Cat#78062) and Fms-like tyrosine kinase-3 Ligand (Flt3, StemCell Technologies, Cat#78009) at 25 ng/mL. On day 5, cells were again collected by gravitation for hydrogel embedding. The hydrogels, composed of 60% collagen and 40% Matrigel, were prepared on ice following the manufacturer's instructions. This included Reduced Growth Factor Matrigel (Corning, Cat#354230) supplemented with 1 mg/mL human collagen type I (Advanced Biomatrix, Cat#5007) and type IV (Advanced Biomatrix, Cat#5022). A 0.5 mL cell-free base layer was first laid and allowed to polymerize for 2 hours, followed by another 0.5 mL layer containing the cell aggregates, also left to polymerize for 2 hours at 37 °C and 5% $CO_2$. The fully polymerized gels with cell aggregates were then supplemented with Phase III media. This media included VEGFA at either 50 ng or 25 ng/mL, VEGFC (where applicable) at 50 or 25 ng/mL, FGF2, BMP4, hSCF, Flt3, Erythropoietin (EPO, StemCell Technologies, Cat#78007), Thrombopoietin (TPO, StemCell Technologies, Cat#78210), Granulocytic Colony-Stimulating Factor (G-CSF, StemCell Technologies, Cat#78012), at 25 ng/mL, and Interleukin-3 (IL3, StemCell Technologies, Cat#78194) and Interleukin-6 (IL6, StemCell Technologies, Cat#78050) at 10 ng/mL. The media was refreshed every 72 hours. On day 12, the organoids were scooped from hydrogels and transferred to 96-well ultralow attachment plates (Thermo Fisher, Cat#174925) for further culturing. The media was refreshed at 1:1 ratio every 72 hours. Organoids are fully matured on day 21 for downstream experiments.

Confocal imaging of the whole-mount organoids was conducted on a Nikon AXR confocal system, utilizing a 20X water immersion objective (CFI Apo LWD Lambda S 20XC WI). Organoids were fixed in 4% PFA and subsequently immunostained with primary and secondary antibodies: anti-human CD45, 1:500 (eBioscience, 14-9457-82); anti-CD140b, 1:200 (Abcam, ab215978),anti-CD71, 1:250 (eBioscience, 14-0719-82); anti-CD235a, 1:200 (PA5-141179); biotin-UEA1, 1:200 (Vector Lab, B0-1065-2), AF647 conjugated goat anti-rabbit IgG (H + L) cross-adsorbed secondary antibody; AF568 conjugated streptavidin; AF488 conjugated goat anti-mouse IgG (H + L) cross-adsorbed secondary antibody. All secondary antibodies were used at 1:300 dilution. Following immunostainings, organoids were dehydrated in ethanol of different concentrations (50%, 70%, 90%, 100%) before tissue clearance with Ethyl Cinnamate and subsequent imaging. Z-stack confocal images were rendered with Imaris 10.0 (Oxford Instruments).

## Engraftment of CD34 + HSPCs
CD34+ hematopoietic stem and progenitor cells (HSPCs) were purchased from StemCell Technologies (Cat#70002). For engraftment, CRISPR/Cas9-edited CD34+ cells were stained with CellVue Claret Far Red Membrane Label (Sigma-Aldrich, Cat#MINCLARET-1KT) following the manufacturer's instructions. $5 \times 10^4$ CellVue-labeled CD34+ cells were suspended in 200 μL StemPro-34 medium (Gibco, Cat#10639011) and combined with 100 μL Matrigel (Corning, Cat#CLS356237) on ice. Organoids on day 18 post-differentiation were deprived of their cell culture medium and exposed to 30 μL of this cell suspension in ultra-low attachment 96-well plates (Thermo Fisher, Cat#174927), followed by a 30-min incubation at 37 °C. Afterward, 150 μL of StemPro-34 medium containing SCF, FLT3, TPO, EPO, and IL3 (10 ng/mL each) was added to support engraftment. The engrafted organoids were eval-

uated via flow cytometry or imaging 3 days after engraftment. We pooled 10 engrafted organoids for each flow cytometry analysis to mitigate the variability between individual organoids.

## Statistics
Results are expressed as mean ± SEM unless otherwise indicated. Statistical comparisons between two groups were performed with two-tailed unpaired Student's t tests, and the comparison among multiple groups was evaluated with 1-way ANOVA tests using GraphPad Prism version 9.0 software. For each experiment, at least three independent biological replicates were tested for each group. Each replicate represents either an individual mouse or a cell culture sample. For Figs. 2M, 5J, and 6O, technical replicates were used due to the difficulty in obtaining fresh patient samples. All technical replicates were independently purified, stained, and tested in separate tubes. For representative data, at least two independent experiments were performed to ensure reproducibility. * $p < 0.05$, ** $p < 0.01$, *** $p < 0.001$, and **** $p < 0.0001$. ns: not significant.

## Reporting summary
Further information on research design is available in the Nature Portfolio Reporting Summary linked to this article.

## Data availability
The CUT&RUN sequencing data are available from the Gene Expression Omnibus (GEO) database under accession code GSE254143. The single-cell sequencing data are available from the GEO database under accession code GSE254144. Raw fastq files are available upon request to the corresponding author. Source data are provided with this paper.

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

## Acknowledgements

We thank Ching Man Wai and Matthew Schipma from the NUSeq Core for their help with the sequencing studies. We thank Susan Ross for providing Ddx41 floxed mice and Nicolae Valentin David for HBBCre mice. This work was supported by the National Institute of Diabetes and Digestive and Kidney Disease (NIDDK) grant R01-DK124220 (P.J.), National Heart, Lung, and Blood Institute (NHLBI) grant R01-HL148012 (P.J.), R01-HL150729 (P.J.), R01-HL169507 (P.J.), R35-HL171168 (L.B.), National Cancer Institute (NCI) grant R00-CA248835 (V.S.). H.B. is a recipient of the F32 Ruth L. Kirschstein Postdoctoral Individual National Research Service Award (F32-HL170648). K.R. is a recipient of the NIH K99 pathway to independent award (K99CA289959) and the EvansMDS foundation Young Investigator award.

## Author contributions

H.B., K.R., P.W., E.L., X.H., J.Y., I.A., K.T., R.M., and Y.T. performed the experiments and interpreted data. H.B., E.T.B., and W.W. performed the CUT&RUN data analyses. H.B., K.R., L.A.G., Y.L., V.S., L.B., M.S., and P.J. analyzed the data. H.B. and P.J. designed the experiments, interpreted data, and wrote the manuscript.

## Competing interests

The authors declare no competing interests.
