## [Transparent Peer Review file · Nature Communications]

DDX41 resolves G-quadruplexes to maintain erythroid genome integrity and prevent cGAS-mediated cell death

Corresponding Author: Dr Peng Ji

Version 0:

Reviewer comments:

Reviewer #1

(Remarks to the Author)

In this manuscript, using mouse models and human organoids, the authors show that DDX41 is essential for regulating erythropoiesis by maintaining G4 homeostasis. The novel findings in this study are the identification of DDX41 interaction with G4 DNA, in addition to the R-Loop structures previously reported by others and the finding that in their embryonic lethal VavCreDDX41 model, that including knockout of cGas rescues the lethality.

Overall, the manuscript is clearly presented and well-organized. However, there are some concerns about the technical aspects of the study which require clarification:

1. The authors report embryonic lethality in VavCre:Ddx41fl/fl mice (Fig. 6B), an observation that differs from a previous study (PMID: 35303436), which indicated survival of VavCre+ Ddx41fl/fl mice survival until after birth. Moreover, the major defect shown previously was in HSPCs and the myeloid lineage with minor effects on the erythroid populations. How do the authors reconcile these differences?
2. VavCre is expressed in HSPCs, so other cell types including macrophages should be deleted for DDX41. Was this seen in the current study?
3. One possible explanation for these differences, which should be discussed, is the different mating setups (males carrying the VavCre gene bred with Ddx41fl/fl females in the previous study; females carrying the Vav-Cre transgene bred with male Ddx41fl/fl mice in the current study). Given that Jackson Lab reports that the VavCre gene is expressed in oocytes, how can the current authors rule out that Ddx41 is not deleted in tissues other than HSPCs and that this contributes to the embryonic lethality? Did they check other tissues for the knockout?
4. A previous study (PMID: 29871919) reported embryonic lethality in germline knockout mice for Ddx41 (CMVCre) but showed that dendritic cell (CD11cCre) and macrophage (LysMCre) tissue-specific knockout mice had no hematopoietic defects and survived normally. This paper should be cited.
5. For experiments carried out with the EpoRCre:Ddx41fl/+ mice, the controls to compare should be EpoRCre, not Ddxfl/fl. The EpoRCre gene is a knockin, so these mice are also heterozygous for the erythropoietin receptor which could play a role in erythroid differentiation.
6. For the studies with the MDS patient derived bone marrow, was the PRPF8 mutation germline or somatic?
7. A previous study (PMID: 35613581) showed that DDX41 has both unwinding and annealing activities, and that the R525H mutant had reduced unwinding activity but no difference in annealing, which seemed to contribute to higher type I interferon production. Do the results of the current study agree with these findings? Please cite this study in the context of the Discussion.
8. Two pathogenic variants of DDX41, G530D and R525H, were investigated. The statement that “the DDX41 mutation is incompatible with erythroid survival” may be too generalized. Using more than one MN patient sample for each mutation, if possible, would strengthen the authors’ point of view. Do the authors predict that all DDX41 mutations or just those that alter

ATP binding will affect erythropoiesis?

9. Several other studies found that loss or mutation of DDX41 leads to aberrant ribosome gene expression/ribosomopathies, albeit in the myeloid lineage not the erythroid lineage. Please reference these studies in the Discussion.
10. DDX41 binds to ssDNA, dsDNA and RNA/DNA hybrids, so binding to G4 DNA is perhaps not unexpected. Is the affinity for G4 DNA higher than for these other molecules?
11. Previous studies have convincingly shown that DDX41 interacts with R-loops. The authors suggest that G4 DNA is related to R-loops. Did they examine R-loops in their study? How do they know that G4 DNA isn't just a "marker" for R-loops in the genome?
12. In Figure 1, the authors demonstrate the phenotype of EpoRCre:Ddx41f/+ and HBBCre:Ddx41f/f to investigate the role of Ddx41 in various stages of erythropoiesis. It would be helpful to show DDX41 expression across different developmental stages.
13. In Figure 5, increased gH2AX was observed in PDS-treated HSPCs, while no increase was seen in erythroblasts from EpoRCre:Ddx41f/+ or HBBCre:Ddx41f/f mice. Have the authors tested G4 levels with Ddx41 deficiency at different stages of erythropoiesis? Does the dependence on G4 vary across different developmental stages?
14. The authors conclude from Supplemental Figures 3-5 that Ddx41 is dispensable for the development of other lineages; However, Ddx41 was only depleted in fully differentiated cells rather than in progenitors.
15. For the scRNA-Seq analysis, the authors need to explain their crosses and what they mean by littermate controls (what is the genotype?). Why didn't they use littermate controls for the other experiments? How old were they mice when they did the analysis? Why would wild type mice have increased cell death, etc.? Moreover, the status of genes involved in cGAS-STING signaling and G-quadruplexes should be included.
16. p. 4 "As reported, Ddx41 deficiency led to embryonic lethality". Reference 12 is incorrect. This manuscript showed that haploinsufficiency coupled with a mutant DDX14 (R525H) led to defects in WBC, RBC and platelets, not embryonic lethality.
17. p. 5 "...which closely mimics patients with DDX41 germline mutations..." Is this statement true – that patients with germline mutation only have anemia and no myelodysplasia before the 2nd hit?
18. The authors find increased G4 in bone marrow mononuclear cells from a MN patient with DDX41 mutation. What was the mutation? Was it somatic or germline? Doesn't this indicate that DDX41 is also affecting the myeloid lineage?
19. Fig. 3H Only colony number and not size are shown.
20. Why do the authors think that HSPCs from the HBBCre:Ddx41f/f mice show decreased rRNA transcription if they don't have any defects in hematopoiesis?
21. Fig. 5I and 5J. Doesn't this suggest that DDX41 mutations affect more than just erythropoiesis, since this analysis was done on total bone marrow?

Reviewer #2

(Remarks to the Author)

Ji et al present novel findings demonstrating that DDX41 as a G4 dissolver, essential for erythroid genome stability and suppressing the cGAS-STING pathway. They show that DDX41 loss leads to erythroid differentiation blocks and the mechanism behind this. These are important studies with translational relevance. They use elegant mouse modeling to demonstrate their mechanism. Overall, a very strong study.

Minor concern:

The western blots may have been done in different gels than the loading controls. Ideally controls can be done on the same gel.

Reviewer #3

(Remarks to the Author)

Reviewer #4

(Remarks to the Author)

In this manuscript Peng and co-worker provide interesting data suggesting that genetic impairment of DDX41 is associated with accumulation of G4s and genomic instability in erythroid. In a nutshell, the authors combine genomics (CUT&RAN) to

map G4s and DDX41 binding sites, biophysics to evaluate DDX41 binding to G4s (and their resolution) and immunofluorescence to visualise co-localization of DDX41 and G4s. Collectively they conclude that impairment of DDX41 leads to accumulation of G4s which leads to DNA damage and eventually cell death under certain conditions.

Whilst the hypothesis is compelling and the authors present some interesting preliminary data in support of their conclusions, this reviewer does not feel that the evidence reported in this manuscript is sufficient to fully support the authors conclusions. In general, it is unclear how the experiments are performed (lack of methods details) and basic standards set by the current state of the art are simply not met. Based on this, I do not feel comfortable supporting publication of this manuscript in Nature Communications.

Here are my main concerns:

1) The G4-mapping is supposedly done using CUT&RAN rather than CUT&Tag. There is no established CUT&RAN protocol for G4-mapping and the reference used by the authors is for a CUT&Tag one. Whilst I appreciate these methods are very similar, this is somehow confusing and the methods do not provide more information. Indeed, it is not even clear what G4-antibody has been used for G4-mapping, I am assuming this is BG4 like for the IF but is not explicitly mentioned in the methods. Additionally, the number of G4s peak detected is suspiciously low. 173 peak or less than 2000 in the best case is not really in line with literature reports and this makes me concerned about the quality of this genomic data. Additional validation with ChIP-Seq (or other complementary methods like Chem-Map) is required to validate such low number of G4 peaks.

2) On the same note of my first concern, the commercial BG4 used for the genomics (I am assuming is the same described in the antibody section of the methods) is not compatible with reliable genomic mapping of G4s. This needs to be performed with in house expressed BG4 to ensure reliable genomic enrichment of G4s, the commercial antibody is only sold for IF applications and is not compatible with genomics (as clearly stated by the provider). This is a polyclonal antibody and the single chain version should be used for ChIP or CUT&Tag.

3) Same concern on the validity of the antibody used for flow cytometry. The commercial BG4 is valid for EMSA, IF and ELISA applications. It is more reasonable to believe it would be suitable for flow-cytometry but again additional validation should be provided.

4) IF experiments seems to show global BG4 and DDX41 staining thorough the nucleus. I don't see the ground on which the authors can talk about co-localization. The quality of the IF in general is very poor and far from the punctuate standards that are typically reported for BG4 staining. Regardless considering the diffuse DDX41 staining I do not believe the authors can invoke any meaningful co-localization via IF

5) The use of FRET for measuring G4-unfolding is very puzzling and not clear to me. Once the G4 is resolved it will quickly refold under the conditions used so this is why these type of assays are never performed in the absence of a complementary DNA sequence that can trap the unfolded G4 and form a duplex. The unfolding data are therefore again not very reliable and do not present sufficient evidence to support unfolding but might simply reflect changes in fluorescence quantum yield when the oligo is bound by the protein. Proper unfolding assays in agreement with the standards of the literature need to be performed. Also, why are these experiments performed without ATP? G4 unfolding should be ATP dependent.

Minor points:

6) The term G4 dissolution is not accurate, the authors should use G4 resolution

7) A paper of 2013 cannot be invoked as recent report

8) The methods sections should be properly populated in the main manuscript, having one line per method is simply not acceptable. Additionally, the lack of methods detail also in the supporting information is not in line with the reproducibility standard expected by any scientific journal

Version 1:

Reviewer comments:

Reviewer #1

(Remarks to the Author)

The authors did a good job of addressing our concerns. The major novel finding in the current manuscript is that DDX41 appears to be important for resolving G4 DNA structures during erythroid development, rather than R loops or transcriptional/translational alterations as has been previously shown by others for multiple hematopoietic lineages.

There are still a few issues that should be addressed:

1. The authors are incorrect about the timing/location of VavCre expression. It is well-documented that the transgene is expressed in hematopoietic stem cells (as well as endothelial cells, among others), and those all lineages derived from these cells should show deletion e.g. see <https://pubmed.ncbi.nlm.nih.gov/12548562/>. So, the authors' response does not make sense. There should be loss of DDX41 in more than just the erythroid lineage, as has been previously reported.

2. Although not a major conclusion of the manuscript, we still do not think that the embryonic lethality is due to knockout of DDX41 in the erythroid lineage in the VavCre mice. The difference between the previous publication showing survival post-birth must be due to the nature of the crosses. In the current manuscript, female VavCre transgenics were crossed with male DDX41^{fl/fl} mice (the previous manuscript did the opposite cross). Since VavCre is expressed in oocytes, as well as in the endothelial lineage, without doing a thorough examination of the entire early embryo, they cannot be sure as to what causes the lethality. The authors should temper their statements about lethality with this cross. In fact, the inclusion of these data is not even necessary to the remainder of the paper, which uses other knockouts that are more specific for the erythroid lineage.

3. Lines 141-142: It has been previously shown that knockout of DDX41 in macrophages and dendritic cells, using the same crosses as reported here, has no effect on cell viability, only on innate immunity (ref. 41). The authors should cite this paper here.

4. line 281. Ref 41 showed that DDX41 bound DNA/RNA hybrids.

Reviewer #2

(Remarks to the Author)

This is a novel translationally important study. The revisions are satisfactory and enhance the manuscript. The study answers some important questions in the field in a rigorous manner.

Reviewer #3

(Remarks to the Author)

Reviewer #4

(Remarks to the Author)

The authors have now addressed my concerns and I am supportive of publication.

We want to thank you and the reviewers for the insightful feedback. We have addressed the reviewers' comments. We apologize for some parts of the manuscript that lack sufficient details, which could lead to misunderstandings from the reviewers. We performed additional studies and have revised the manuscript thoroughly as per the reviewers' suggestions and comments. Please find our point-by-point response to the reviewers' critiques below. Sections modified are highlighted in yellow in the revised manuscript. Notably, we moved the methods section from the supplemental materials to the main text. A new author was also included in the revised manuscript.

Reviewer #1:

1. The authors report embryonic lethality in VavCre:Ddx41^{fl/fl} mice (Fig. 6B), an observation that differs from a previous study (PMID: 35303436), which indicated survival of VavCre+ Ddx41^{fl/fl} mice survival until after birth. Moreover, the major defect shown previously was in HSPCs and the myeloid lineage with minor effects on the erythroid populations. How do the authors reconcile these differences?

Response: We appreciate the reviewer's thoughtful comment and the opportunity to clarify this point. The embryonic lethality phenotype in these mice is consistent with a recently published report (PMID: 38937548) and a previous report (PMID 34473945). We are not sure why the mice could survive to birth in the study the reviewer referred to (PMID: 35303436). However, it is critical to perform Western blotting to ensure the complete depletion of the protein. This paper (PMID: 35303436) only provided genotyping results. It is likely that the small amount of Ddx41 protein sustained the survival of these mice. In our manuscript, we ensure the complete knockout of Ddx41 through genotyping and Western blotting.

Similarly, it is unclear why this paper (PMID: 35303436) showed defects in HSPCs and myeloid lineages and minor effects on the erythroid lineage. It could be again due to the residual Ddx41 proteins. However, similar erythroid defects were clearly documented in the same published report (PMID: 38937548) mentioned above.

2. VavCre is expressed in HSPCs, so other cell types including macrophages should be deleted for DDX41. Was this seen in the current study?

Response: Thank you for this important question. VavCre expression begins around embryonic day 13.5 (E13.5). In contrast, macrophage differentiation starts earlier during development, and tissue-resident macrophage populations are established before E13.5 (PMID: 26681456, 16020514, 2358750, 6883456). As such, these early macrophages may escape Ddx41 deletion in the VavCre model, and their development may not be significantly impacted.

To directly investigate the role of DDX41 in macrophages, we examined LysMCre:Ddx41^{fl/fl} mice, in which Ddx41 is specifically deleted in monocytes and macrophages. Notably, these

mice did not exhibit any overt phenotype, suggesting that Ddx41 is not essential for macrophage maintenance or function in this context. These findings together suggest that the embryonic lethality observed in the VavCre model is unlikely to be driven by defects in the macrophage lineage.

3. One possible explanation for these differences, which should be discussed, is the different mating setups (males carrying the VavCre gene bred with Ddx41^{fl/fl} females in the previous study; females carrying the Vav-Cre transgene bred with male Ddx41^{fl/fl} mice in the current study). Given that Jackson Lab reports that the VavCre gene is expressed in oocytes, how can the current authors rule out that Ddx41 is not deleted in tissues other than HSPCs and that this contributes to the embryonic lethality? Did they check other tissues for the knockout?

Response: We appreciate the reviewer raising this possibility. In our study, we used female mice carrying the VavCre transgene crossed with male Ddx41^{fl/fl} mice. While Jackson Laboratory reported possible expression of VavCre in oocytes, if significant recombination occurred at the oocyte stage, we would expect global deletion of Ddx41, leading to early embryonic lethality before implantation or a complete failure to recover embryos, neither of which we observed.

Importantly, we collected embryos at E13.5, indicating that any potential oocyte-mediated recombination is minimal or negligible. Additionally, the observed hematopoietic-specific defects and embryonic lethality are consistent with VavCre's known activation window starting around E13.5 and support the conclusion that Ddx41 deletion is largely restricted to the hematopoietic compartment.

4. A previous study (PMID: 29871919) reported embryonic lethality in germline knockout mice for Ddx41 (CMVCre) but showed that dendritic cell (CD11cCre) and macrophage (LysMCre) tissue-specific knockout mice had no hematopoietic defects and survived normally. This paper should be cited.

Response: We thank the reviewer for pointing out this relevant study. We have now cited this study in the revised manuscript and incorporated it into the discussion to provide a broader context for our findings and to highlight the consistency across models using different Cre drivers.

5. For experiments carried out with the EpoRCre:Ddx41^{fl/+} mice, the controls to compare should be EpoRCre, not Ddx^{fl/fl}. The EpoRCre gene is a knockin, so these mice are also heterozygous for the erythropoietin receptor which could play a role in erythroid differentiation.

Response: We thank the reviewer for pointing out this important consideration regarding the use of appropriate controls for EpoRCre-mediated deletion. As shown below, we compared EpoRCre-carrying mice with wild-type controls, and performed complete blood count analysis. This comparison showed no significant differences in different parameters, suggesting that the EpoRCre allele does not independently affect erythroid differentiation under steady-state conditions. These findings indicate that the phenotypes observed in our EpoRCre:Ddx41^{fl/fl} mice are due to loss of Ddx41, rather than any confounding effects of the EpoRCre knock-in allele itself.

6. For the studies with the MDS patient derived bone marrow, was the *PRPF8* mutation germline or somatic?

Response: Based on follow-up sequencing after treatment, the *PRPF8* mutation was no longer detectable, suggesting it was most likely a somatic mutation. For clarity, we have now included this information in the revised manuscript.

7. A previous study (PMID: 35613581) showed that *DDX41* has both unwinding and annealing activities, and that the R525H mutant had reduced unwinding activity but no difference in annealing, which seemed to contribute to higher type I interferon production. Do the results of the current study agree with these findings? Please cite this study in the context of the Discussion.

Response: Yes, our results are consistent with the findings reported in PMID: 35613581. In our FRET-based assay, the R525H mutant of *DDX41* also showed a loss of unwinding activity, supporting the idea that this mutation impairs helicase function. We have now cited this study and incorporated a brief discussion of these mechanistic parallels in the revised manuscript.

8. Two pathogenic variants of *DDX41*, G530D and R525H, were investigated. The statement that “the *DDX41* mutation is incompatible with erythroid survival” may be too generalized. Using more than one MN patient sample for each mutation, if possible, would strengthen the authors’ point of view. Do the authors predict that all *DDX41* mutations or just those that alter ATP binding will affect erythropoiesis?

Response: We appreciate the reviewer’s point and agree that the statement may be overly broad. We have revised the language in the results and discussion sections of the manuscript to be more specific.

Currently, we do not know whether all *DDX41* mutations affect erythropoiesis to the same extent. However, based on our current findings, and consistent with a previous report (PMID: 31548374, reference 18 in the manuscript), the ability of *DDX41* to resolve G-quadruplex (G4) structures does not appear to depend on ATP binding or hydrolysis, suggesting that mutations affecting G4 resolution (even those outside the ATP-binding domain) may still compromise

erythropoiesis. Nevertheless, additional functional studies across a broader range of DDX41 mutations will be necessary to fully address this question.

9. Several other studies found that loss or mutation of DDX41 leads to aberrant ribosome gene expression/ribosomopathies, albeit in the myeloid lineage not the erythroid lineage. Please reference these studies in the Discussion.

Response: We thank the reviewer's suggestion and have included these studies in the revised manuscript in the discussion section.

10. DDX41 binds to ssDNA, dsDNA and RNA/DNA hybrids, so binding to G4 DNA is perhaps not unexpected. Is the affinity for G4 DNA higher than for these other molecules?

Response: We thank the reviewer for this insightful point. Indeed, DDX41 has been previously reported to bind to a variety of nucleic acid structures. In our study, we specifically assessed the binding affinity of DDX41 to G4 DNA structures and found that its affinity for G4 DNA is notably higher than for non-G4 control oligonucleotides of similar sequence composition (**Figure 4D**).

In the revised manuscript, we performed a direct side-by-side comparison with G4s, unfolded G4s, ssDNA, dsDNA, and RNA/DNA hybrids. Our data suggest that G4 DNA represents a preferred or high-affinity substrate for DDX41 compared to other molecules. This is consistent with the idea that DDX41 may have evolved a specialized role in resolving G-quadruplexes, which pose unique challenges to genome stability. This result was presented in the **new Figure 4J**.

11. Previous studies have convincingly shown that DDX41 interacts with R-loops. The authors suggest that G4 DNA is related to R-loops. Did they examine R-loops in their study? How do they know that G4 DNA isn't just a "marker" for R-loops in the genome?

Response: As we mentioned above in response to the reviewer's comment #10, we have compared the affinities of DDX41 with different nucleic acids, including DNA/RNA hybrids. The findings suggest that DDX41 binds more preferentially to G4 than other molecules (**Figure 4J**).

Several lines of evidence also suggest that the effects we observe are primarily driven by G4 accumulation rather than G4s simply serving as a marker for R-loops: 1. G4-selective staining and chemical probing (using BG4 antibody and G4-stabilizing ligands like PDS) revealed a robust increase in G4 levels in various conditions. 2. Treatment with PDS, which stabilizes G4 structures independently of R-loops, phenocopied the erythroid differentiation defects observed with DDX41 loss.

12. In Figure 1, the authors demonstrate the phenotype of EpoRCre:Ddx41f/+ and

HBBCre:Ddx41^{fl/fl} to investigate the role of Ddx41 in various stages of erythropoiesis. It would be helpful to show DDX41 expression across different developmental stages.

Response: We agree with the reviewer that examining DDX41 expression across erythroid developmental stages is important for interpreting the phenotypes observed in the EpoRCre and HBBCre models. In the revised manuscript, we have now included Western blotting analysis of cultured mouse bone marrow erythroid cells at different maturation stages. These data show that Ddx41 is highly expressed during early erythroid differentiation (day 0 to day 1) and declines as cells progress toward terminal maturation (day 2), which supports our model that DDX41 is particularly critical during the early stages of erythropoiesis. We have added these results to the revised **new Figure 1A** and updated the corresponding text in the Results section.

13. In Figure 5, increased gH2AX was observed in PDS-treated HSPCs, while no increase was seen in erythroblasts from EpoRCre:Ddx41^{fl/+} or HBBCre:Ddx41^{fl/fl} mice. Have the authors tested G4 levels with Ddx41 deficiency at different stages of erythropoiesis? Does the dependence on G4 vary across different developmental stages?

Response: We thank the reviewer for this insightful question. We have examined G4 levels at different stages of erythroid differentiation, as shown in the original Figure 2 in the wild-type erythroid cells. In that experiment, we cultured erythroid cells for 2 days in Epo medium and observed that G4 levels were significantly elevated on day 1, corresponding to the peak of proliferation and DNA replication. By day 2, when over 30% of cells were enucleated, G4 levels were markedly reduced. These findings were consistent in mouse and human erythroid cultures, suggesting that G4 accumulation is tightly linked to proliferative stages of erythropoiesis.

To address the reviewer's question directly, we examined G4 levels in erythroid cells from EpoRCreDdx41^{fl/fl} mice, as shown in **original Figure 2F-H**, and HBBCreDdx41^{fl/fl} mice, as shown in the **original Figure 2I**. These data demonstrate that G4 levels were further elevated in the early stage of terminal erythropoiesis in Ddx41-deficient erythroid cells compared to controls, supporting the idea that DDX41 plays an important role in regulating G4 structures during the highly proliferative phase of erythropoiesis.

Additionally, to validate our observations, we repeated the experiment using an alternative method involving an electrochemical probe for G4 detection, which produced consistent results. These confirmatory data are now included in the revised manuscript as **new Supplementary Figures 6A and 6B**. Together, these findings indicate that G4 accumulation is both stage-specific and Ddx41-dependent, with the greatest sensitivity during early erythroid differentiation when replication stress is highest.

14. The authors conclude from Supplemental Figures 3-5 that Ddx41 is dispensable for the development of other lineages; However, Ddx41 was only depleted in fully differentiated cells rather than in progenitors.

Response: We agree that these results primarily demonstrate that Ddx41 is not essential for the maintenance or function of mature cells in these lineages under steady-state conditions.

To reflect this more accurately, we have revised the relevant statements in the revised manuscript to clarify that our data suggest Ddx41 is dispensable for the mature forms of these lineages, and that additional models targeting progenitor stages would be required to fully assess its role during lineage commitment and development.

15. For the scRNA-Seq analysis, the authors need to explain their crosses and what they mean by littermate controls (what is the genotype?). Why didn't they use littermate controls for the other experiments? How old were they mice when they did the analysis? Why would wild type mice have increased cell death, etc.? Moreover, the status of genes involved in cGAS-STING signaling and G-quadruplexes should be included.

Response: We thank the reviewer for these thoughtful and important questions. For the scRNA-Seq analysis, we used total bone marrow cells from Ddx41^{fl/fl} and VavCre Ddx41^{fl/fl}cGas^{-/-} derived from the same litters. The genotypes of these mice were labeled in the figures. We have now provided a detailed breeding strategy in the revised Methods section.

Regarding the use of littermate controls in other experiments, we used littermates whenever feasible, especially in embryonic and early postnatal studies. We have now provided more detailed descriptions of genotypes and control strategies throughout the revised manuscript to improve clarity and transparency.

The scRNA-Seq was performed on 2-month-old mice, as indicated in the figure legend. DKO cells show decreased apoptosis because late-stage erythroblasts have increased apoptotic signals, although they do not undergo apoptosis. These signals are important for chromatin condensation, as previously reported (PMID: 11208865, 17905508, etc.). It is likely that DKO cells have altered chromatin condensation due to the reduction of these signals. We have included a brief discussion about this in the revised manuscript.

We appreciate the reviewer's suggestion regarding the cGAS-STING pathway. As cGas is genetically deleted in the DKO mice, we expect downstream transcriptional activation (e.g., type I interferon-stimulated genes) to be significantly attenuated, and thus the scRNA-Seq data are not suitable for assessing canonical cGAS-STING activation. Nevertheless, we did examine the baseline expression of key genes in this pathway (e.g., *Sting1*, *Tbk1*, *Irf3*, and *Ifnb1*) across genotypes. These genes were either lowly expressed or unchanged across major hematopoietic lineages, consistent with the genetic inactivation of cGas and the absence of robust interferon signaling.

16. p. 4 "As reported, Ddx41 deficiency led to embryonic lethality". Reference 12 is incorrect. This manuscript showed that haploinsufficiency coupled with a mutant DDX14 (R525H) led to defects in WBC, RBC and platelets, not embryonic lethality.

Response: We changed this reference to a more recent one that showed embryonic lethality using VavCre model (PMID 38937548).

17. p. 5 “...which closely mimics patients with *DDX41* germline mutations...” Is this statement true – that patients with germline mutation only have anemia and no myelodysplasia before the 2nd hit?

Response: We included more references regarding the clinical manifestations of patients with *DDX41* germline mutations before developing into full MNs. Specifically, below are some of the direct quotations from the references. Since this model has *DDX41* heterozygosity only in the erythroid lineage, we revised the sentence to “partially mimics patients with *DDX41* germline mutations...”

*“The majority of carriers unaffected by or prior to HMs had normal peripheral blood counts well into adulthood (9 of 15 [60%]; see premalignancy characteristics in Table 1), suggesting that haploinsufficiency of *DDX41* is sufficient for normal baseline hematopoiesis. Cytopenias or macrocytosis (mean corpuscular volume >100 fL) developed in the remaining 6 at a mean age of 66 years (range, 50-85 years) and led to an HM diagnosis shortly thereafter in.”*

<https://pmc.ncbi.nlm.nih.gov/articles/PMC4968341/#:~:text=hematopoiesis,C%29.%20Germ%20line>

*“At first diagnosis of myeloid disorder, according to the World Health Organization classification, 4 patients had MDS with multilineage dysplasia, 11 had MDS with excess of blasts, 11 had AML (M2, M1, and M0; no M612), 3 had MDS/myeloproliferative neoplasm, 3 had aplastic anemia, and 1 had isolated neutropenia (Table 1). Importantly, 15 patients (46%) had a previous history of cytopenia, starting a median of 5.2 years (range, 1.8-15.2 years) before diagnosis of myeloid malignancy. This contrasts with the initial report from Lewinsohn et al,⁷ where germline *DDX41* mutation carriers had normal blood counts until overt myeloid malignancy.”*

<https://ashpublications.org/blood/article/134/17/1441/374965/Germline-DDX41-mutations-define-a-significant>

18. The authors find increased G4 in bone marrow mononuclear cells from a MN patient with *DDX41* mutation. What was the mutation? Was it somatic or germline? Doesn't this indicate that *DDX41* is also affecting the myeloid lineage?

Response: We thank the reviewer for this question. As stated in the manuscript on page 6, the mutation in this patient is a somatic *DDX41* G530D mutation. This is the same patient featured in Figure 1K.

As discussed in the Discussion section, we do not rule out the possibility that *DDX41* deficiency may impact the myeloid lineage, particularly with aging and persistent G4 accumulation. While our current study focuses primarily on the erythroid lineage, which is more sensitive to increased G4, the elevated G4 levels observed in the patient's bone marrow mononuclear cells support the idea that genomic instability due to unresolved G4 structures may eventually contribute to dysfunction in other hematopoietic compartments, including the myeloid lineage.

19. Fig. 3H Only colony number and not size are shown.

Response: We have removed the statement about size in the sentence.

20. Why do the authors think that HSPCs from the $HBBCre:Dcx41^{fl/fl}$ mice show decreased rRNA transcription if they don't have any defects in hematopoiesis?

Response: To clarify, we did not observe ribosome defects in HSPCs from $HBBCre:Ddx41^{fl/fl}$ mice. Instead, we find decreased ribosomal protein levels in late-stage erythroid cells, which correlate with Ddx41 loss at this stage of erythropoiesis in these mice. In Figure 5F, HSPCs were the progenitor cells at the beginning of the culture. On Day 1 of the culture, these cells were all differentiated into erythroid. And these erythroid cells were the cell type we used to assess rRNA transcription.

21. Fig. 5I and 5J. Doesn't this suggest that DDX41 mutations affect more than just erythropoiesis, since this analysis was done on total bone marrow?

Response: Our study primarily focuses on the erythroid lineage, which has a stronger phenotype and is more sensitive to G4 elevation. However, we agree that these data suggest DDX41 mutations may have broader effects beyond erythropoiesis, consistent with its known roles in genome stability and nucleic acid sensing. We have included more discussion in the revised manuscript.

Reviewer #2:

Minor concern:

The western blots may have been done in different gels than the loading controls. Ideally controls can be done on the same gel.

Responses: We appreciate the reviewer's general positive comments and attention to the details. In several experiments, we analyzed multiple target proteins, which made it technically challenging to run all blots, including loading controls, on the same gel and membrane. In such cases, loading controls were run on parallel gels using the same lysates, under identical conditions.

However, in experiments where fewer target proteins were analyzed, we did run the loading control and target protein on the same gel, as the reviewer suggested. We have now clarified this in the revised figure legends, and we remain committed to presenting data with accurate and appropriate loading controls.

Reviewer #4:

Whilst the hypothesis is compelling and the authors present some interesting preliminary data in support of their conclusions, this reviewer does not feel that the evidence reported in this manuscript is sufficient to fully support the authors conclusions. In general, it is unclear how the experiments are performed (lack of methods details) and basic standards set by the current state of the art are simply not met. Based on this, I do not feel comfortable supporting publication of this manuscript in Nature Communications.

Response: We appreciate the reviewer's feedback and the acknowledgment that our hypothesis is compelling. In the revised manuscript, we have further expanded the Methods section to improve clarity and ensure that all experimental procedures are described with sufficient detail for reproducibility. We hope this addition helps address the reviewer's concern regarding methodological transparency and rigor.

Main concerns:

1) The G4-mapping is supposedly done using CUT&RAN rather than CUT&Tag. There is no established CUT&RAN protocol for G4-mapping and the reference used by the authors is for a CUT&Tag one. Whilst I appreciate these methods are very similar, this is somehow confusing and the methods do not provide more information. Indeed, it is not even clear what G4-antibody has been used for G4-mapping, I am assuming this is BG4 like for the IF but is not explicitly mentioned in the methods. Additionally, the number of G4s peak detected is suspiciously low. 173 peak or less than 2000 in the best case is not really in line with literature reports and this makes me concerned about the quality of this genomic data. Additional validation with ChIP-Seq (or other complementary methods like Chem-Map) is required to validate such low number of G4 peaks.

Response: In our original study, we indeed used a CUT&RUN-based protocol rather than CUT&Tag. We apologize for the confusion in the original manuscript. The detailed method for the revised G4 mapping methodology is included in the revised manuscript.

Regarding the antibody used, we confirm that the G4 mapping experiments were performed using the BG4 antibody from Absolute Antibody (catalog no. Ab00174-23.0), consistent with our immunofluorescence studies. We have now clearly stated this in the revised Methods section.

To address the concern about the relatively low number of detected G4 peaks, we conducted additional experiments using two antibodies: The BG4 antibody from Absolute Antibody (used

in the original manuscript), and the Millipore Sigma BG4 antibody (MABE917), which is advertised for ChIP applications.

We performed ChIP-seq and repeated CUT&RUN using these antibodies during the revision of the manuscript. We found that ChIP-seq yielded substantially fewer erythroid-specific peaks and higher background noise compared to CUT&RUN (See data below). This suggests that ChIP-seq is suboptimal for G4 detection in erythroid cells, likely due to the inherent fragility of erythroid chromatin and the harsh chromatin shearing steps required for ChIP. Erythroid chromatin is highly condensed and sensitive to disruption, which may compromise the detection of G4 structures when using ChIP-seq. In contrast, CUT&RUN, which uses gentle in situ enzymatic cleavage without harsh sonication, preserves native DNA structures better, making it a more suitable method for G4 detection in erythroid cells. Therefore, we presented only the CUT&RUN data in the revised manuscript.

Regarding the low peak numbers, our peak-calling strategy in the original CUT&RUN analysis employed an extremely stringent cutoff ($p < 0.0001$), optimized for high-confidence peak identification. Upon reanalyzing the CUT&RUN data with standard parameters, we now identify 1,088 G4 peaks in HSPCs and 2,745 G4 peaks in erythroid cells (**new Figure 4A**), consistent with literature expectations. Thus, the apparent low peak numbers in the original submission were due to a deliberately conservative analysis approach, rather than an inherent technical limitation.

This revised analysis prompted us to refine our original conclusion. While we initially reported an increase in Ddx41 and G4 colocalization during differentiation, the expanded dataset reveals that the proportion of overlapping peaks remains relatively stable, even though the total number of Ddx41 and G4 peaks increases markedly as cells differentiate into erythroid cells (**new Figure 4B**).

Further analysis of the full peak set revealed that although the majority of G4 peaks are located in intergenic regions, both Ddx41 and G4 peaks show enrichment at promoter regions. Notably,

this promoter enrichment becomes more pronounced as differentiation progresses toward the erythroid lineage (**new Figure 4A**).

2) On the same note of my first concern, the commercial BG4 used for the genomics (I am assuming is the same described in the antibody section of the methods) is not compatible with reliable genomic mapping of G4s. This needs to be performed with in house expressed BG4 to ensure reliable genomic enrichment of G4s, the commercial antibody is only sold for IF applications and is not compatible with genomics (as clearly stated by the provider). This is a polyclonal antibody and the single chain version should be used for CHIP or CUT&Tag.

We appreciate the reviewer's point about antibody source and specificity. In the original manuscript, we used the commercial BG4 antibody from Absolute Antibody (catalog no. Ab00174-23.0) that was validated for IF applications but not explicitly for genomic assays. To address this, we performed comparative ChIP-Seq experiments in HSPCs using the BG4 antibody from Absolute Antibody and the Millipore BG4 (MABE917) antibody validated for ChIP-Seq. The Millipore antibody detects more peaks overall. The peaks identified by the Absolute antibody, although much less, largely overlapped with those detected by the Millipore antibody (See data below on the left). We also performed a principal component analysis (PCA) using peak locations, which revealed that the Millipore antibody produced substantially higher variability among replicates compared to the Absolute antibody, suggesting that the Absolute antibody may have greater specificity under our experimental conditions (see data below on the right).

As mentioned above, we repeated CUT&RUN analyses using these two antibodies in both HSPCs and erythroid cells. PCA analysis samples processed with the Absolute antibody clustered distinctly by cell type, with HSPCs and erythroid cells forming two separate groups. In contrast, samples processed with the Millipore antibody displayed an abnormal clustering pattern: two HSPC replicates clustered with erythroid cells, while the third HSPC replicate was an outlier, positioned far from all other samples (See data below).

Thus, under our experimental conditions, the Absolute Antibody BG4 reagent provided more reproducible and biologically meaningful enrichment patterns, particularly when applied to fragile erythroid chromatin using CUT&RUN. We therefore believe its use is justified for the specific context of this study.

3) Same concern on the validity of the antibody used for flow cytometry. The commercial BG4 is valid for EMSA, IF and ELISA applications. It is more reasonable to believe it would be suitable for flow-cytometry but again additional validation should be provided.

We acknowledge the reviewer's point that the BG4 antibody is not officially validated for flow cytometry applications. Although the Absolute Antibody BG4 product is marketed primarily for IF, ELISA, and EMSA, we reasoned that flow cytometry involves a detection mechanism similar to IF, antigen recognition under native or lightly fixed conditions, and thus could be feasible.

To provide further validation, we confirmed that BG4-positive flow cytometry signals were specific and sensitive to treatments known to induce G4 upregulation (e.g., treatment of PDS in Figure 3). Moreover, control experiments using isotype-matched antibodies showed minimal background staining. To further support our conclusion, we also employed N-Methyl Mesoporphyrin IX (NMM/NMM580), a widely used chemical probe for G4 detection. As shown in Supplemental Figure 6, flow cytometry using NMM staining produced results consistent with those obtained using the BG4 antibody. Together, these experiments support that the observed flow cytometry signals reflect bona fide G4 structures.

4) IF experiments seems to show global BG4 and DDX41 staining through the nucleus. I don't see the ground on which the authors can talk about co-localization. The quality of the IF, in general, is very poor and far from the punctuate standards that are typically reported for BG4

staining. Regardless, considering the diffuse DDX41 staining, I do not believe the authors can invoke any meaningful co-localization via IF.

Response: We thank the reviewer for their feedback regarding the immunofluorescence (IF) data quality and interpretation. In our initial submission, both BG4 and Ddx41 staining appeared diffuse across the nucleus, which contrasts with the punctate nuclear foci sometimes reported in other systems, particularly in transformed or adherent cell lines.

To address this concern, we repeated the experiments using confocal immunofluorescence microscopy and incorporated a FLAG-tagged BG4 antibody construct to improve detection specificity and resolution. Despite these efforts, the staining patterns for both BG4 and endogenous Ddx41 in primary hematopoietic cells remained predominantly diffuse rather than punctate (**new Figure 4C**). This likely reflects inherent biological differences in chromatin structure and nuclear organization in these primary, non-adherent cells. Notably, previous studies describing punctate BG4 staining have largely focused on transformed or immortalized cell lines, where G-quadruplex structures may be more stably folded or enriched at specific genomic loci.

Importantly, while the diffuse nuclear distribution precludes high-confidence assertions about discrete subnuclear co-localization, the spatial overlap in their nuclear presence is supported by our CUT&RUN data, which demonstrate significant genomic co-enrichment of Ddx41 and G4 peaks (**new Figure 4A-B**). We have therefore revised the manuscript to describe the IF findings as consistent with general nuclear co-distribution.

5) The use of FRET for measuring G4-unfolding is very puzzling and not clear to me. Once the G4 is resolved it will quickly refold under the conditions used so this is why these type of assays are never performed in the absence of a complementary DNA sequence that can trap the unfolded G4 and form a duplex. The unfolding data are therefore again not very reliable and do not present sufficient evidence to support unfolding but might simply reflect changes in fluorescence quantum yield when the oligo is bound by the protein. Proper unfolding assays in agreement with the standards of the literature need to be performed. Also, why are these experiments performed without ATP? G4 unfolding should be ATP dependent.

We appreciate the reviewer's thoughtful comments regarding the interpretation of our FRET-based G4 unfolding assays. Regarding the concern that unfolded G4 structures may spontaneously refold under our assay conditions, we would like to clarify that the FRET assay was carefully designed to account for this possibility. The G4 oligonucleotides were initially folded by heating at 95°C in a high-concentration potassium (K⁺) folding buffer, followed by slow cooling to promote stable G4 structure formation. However, the FRET-based unfolding assays were subsequently conducted in a low-K⁺ unfolding buffer, which substantially reduces the thermodynamic stability of G4 structures and minimizes spontaneous refolding. As a control, we included unfolded oligos that were prepared in the same unfolding buffer but without the folding step. The FRET signals from these unfolded controls confirmed that under the assay

conditions, the oligos do not re-establish stable G4 structures spontaneously (Supplementary Figure 8C). Thus, the observed FRET signal changes upon DDX41 treatment are attributable to the destabilization or resolution of pre-folded G4 structures, rather than simple alterations in fluorescence quantum yield due to protein binding.

Regarding the role of ATP, our FRET assay followed the standard approach established in previous studies of DDX family helicases, particularly DDX5 (ref 18 in our manuscript), which showed that G4 resolution can occur independently of ATP hydrolysis. In our experiments, DDX41 similarly promoted G4 unfolding in the absence of ATP, consistent with a mechanism in which DDX41 binds and destabilizes G4 structures without requiring ATP for the initial resolution step. ATP is typically required for helicase dissociation from nucleic acid substrates after remodeling, not for the initial structural destabilization. Indeed, in our assays, the unfolding reaction reached a rapid saturation point, supporting that DDX41 activity is ATP-independent for G4 resolution under these conditions.

We have now clarified these experimental details and interpretations in the revised manuscript to address the reviewer's important points.

Minor points:

6) The term G4 dissolution is not accurate, the authors should use G4 resolution

Response: This was changed.

7) A paper of 2013 cannot be invoked as recent report

Response: This was changed.

8) The methods sections should be properly populated in the main manuscript, having one line per method is simply not acceptable. Additionally, the lack of methods detail also in the supporting information is not in line with the reproducibility standard expected by any scientific journal

Response: Thank you for pointing this out. We have significantly expanded the Methods section in the main manuscript to include detailed descriptions of major experimental and analytical procedures to ensure clarity and reproducibility. We have also added further methodological details to the Supporting Information, which is in line with the journal's standards. We appreciate the reviewer's emphasis on transparency and have revised accordingly.

We would like to thank you again for the favorable review of our manuscript titled “DDX41 resolves G-quadruplexes to maintain erythroid genome integrity and prevent cGAS-mediated cell death” (NCOMMS-24-66795A-Z). We appreciate the constructive feedback provided by you and the reviewers, which has helped us improve the clarity and rigor of our work. Below, we provide detailed point-by-point responses to the reviewers’ comments. The manuscript has been revised accordingly and follows the author checklist. All changes have been tracked in the revised document for your review.

Reviewer #1 (Remarks to the Author):

The authors did a good job of addressing our concerns. The major novel finding in the current manuscript is that DDX41 appears to be important for resolving G4 DNA structures during erythroid development, rather than R loops or transcriptional/translational alterations as has been previously shown by others for multiple hematopoietic lineages.

There are still a few issues that should be addressed:

1. The authors are incorrect about the timing/location of VavCre expression. It is well-documented that the transgene is expressed in hematopoietic stem cells (as well as endothelial cells, among others), and those all lineages derived from these cells should show deletion e.g. see <https://pubmed.ncbi.nlm.nih.gov/12548562/>. So, the authors’ response does not make sense. There should be loss of DDX41 in more than just the erythroid lineage, as has been previously reported.

We thank the reviewer for this important clarification. We agree that VavCre is active in hematopoietic stem cells and endothelial cells, as previously reported. Our original statement was intended to describe the erythroid-dominant phenotype observed upon Ddx41 deletion, not to imply lineage-restricted Cre activity. We have revised the text to more accurately reflect the known expression pattern of VavCre.

2. Although not a major conclusion of the manuscript, we still do not think that the embryonic lethality is due to knockout of DDX41 in the erythroid lineage in the VavCre mice. The difference between the previous publication showing survival post-birth must be due to the nature of the crosses. In the current manuscript, female VavCre transgenics were crossed with male DDX41^{fl/fl} mice (the previous manuscript did the opposite cross). Since VavCre is expressed in oocytes, as well as in the endothelial lineage, without doing a thorough examination of the entire early embryo, they cannot be sure as to what causes the lethality. The authors should temper their statements about lethality with this cross. In fact, the inclusion of these data is not even necessary to the remainder of the paper, which uses other knockouts that are more specific for the erythroid lineage.

We appreciate the reviewer’s thoughtful point. We agree that the embryonic lethality observed in VavCre:Ddx41^{fl/fl} mice may involve contributions from non-hematopoietic lineages, as VavCre is also expressed in oocytes and endothelial cells. We have revised the text accordingly to acknowledge this possibility and have tempered our interpretation of the lethality.

3. Lines 141-142: *It has been previously shown that knockout of DDX41 in macrophages and dendritic cells, using the same crosses as reported here, has no effect on cell viability, only on innate immunity (ref. 41). The authors should cite this paper here.*

We appreciate the reviewer's suggestion. We have already cited reference 41 (Stavrou et al., *mBio*, 2018) in the **Discussion** section in the last revision to contextualize our findings with prior work showing that DDX41 knockout in dendritic cells and macrophages does not impair viability. This study supports our observation that Ddx41 is dispensable for cell survival in multiple differentiated myeloid lineages.

4. line 281. *Ref 41 showed that DDX41 bound DNA/RNA hybrids.*

We appreciate this correction. We have revised the text to reflect that reference 41 (Stavrou et al., *mBio*, 2018) showed that DDX41 binds to DNA/RNA hybrids. This clarification has been incorporated into the revised manuscript at the relevant location to accurately describe the known nucleic acid-binding properties of DDX41.